# Inter-Domain Sensor Alignment for Unsupervised Domain Adaptation of Wearable Multivariate Time Series

## Abstract

Unsupervised domain adaptation (UDA) for multivariate time-series (MTS) data in the wearable domain transfers knowledge from a labeled source to an unlabeled target, typically with signals collected from multiple body-worn sensors. Although existing UDA methods devote substantial effort to modeling temporal shifts, they often rely on simple spatial alignment across domains, thereby limiting their capacity for effective adaptation. Real systems in the wearable domain exhibit *sensor-wise domain shift*, including changes in placement or orientation, which necessitates the explicit consideration of inter-domain spatial sensor relations. Therefore, we introduce **IDSA**, *Inter-Domain Sensor Alignment for wearable MTS-UDA*, a plug-in module that augments any base UDA loss with two complementary components: (i) an *inter-domain sensor transport* that learns a cross-sensor relation matrix from domain-specific sensor embeddings and transports target channels toward the source, and (ii) a *channel decorrelation* regularizer that sparsifies intra-domain graphs to suppress redundant or noisy couplings. Our sensor transportation loss is shown to be equivalent (up to a constant) to the discrete 1-Wasserstein objective. When used as a plug-in with Deep CORAL or CLUDA, IDSA achieves consistent gains across five HAR and sEMG benchmarks compared to recent baselines in activity classification accuracy, achieving a performance enhancement in most scenarios.

## 1 Introduction

Multivariate time-series sensor data play a vital role in the wearable domain, where multi-sensor signals enable tasks such as human activity recognition (Zhong et al., 2022) and gesture classification (Tchantchane et al., 2023). These tasks support high-impact applications in health monitoring (Jijesh et al., 2021), rehabilitation (Schrader et al., 2020), and human-computer interaction (Qi et al., 2019), highlighting the practical importance of achieving reliable performance on wearable sensor data. Body-worn devices record concurrent signals from multiple sensors, yielding data with rich spatial dependencies between sensors as well as temporal dynamics. This inherent spatio-temporal complexity gives rise to substantial distribution shifts, leading to significant discrepancies across different domains, such as variations among subjects or across repeated measurements (Wang et al., 2023). In practice, changes in placement, orientation of sensors induce *sensor-wise domain shift* across different domains, which is distinct from the usual temporal covariate shift and remains relatively underexplored. (Banos et al., 2014)

Unsupervised domain adaptation (UDA) has been widely explored to enhance model performance under domain shift (Huang et al., 2022; Ozyurt et al., 2023; Li et al., 2022), as an effective approach to resolve this problem. The goal of UDA is to transfer knowledge from a source domain to a target domain without access to labels in the target domain dataset, thereby saving annotation costs in the target domain, and has shown significant performance improvements in various fields (Rangwani et al., 2022; Wei et al., 2024) and wearable applications (Ozyurt et al., 2023) in particular.

Existing UDA methods have devoted significant effort to modeling distribution shifts along the temporal dimension, leading to performance gain for UDA on multivariate time series (MTS) data (He et al., 2023; Liu et al., 2024). However, they leave the *inter-domain multi-sensor spatial structure ei-*

Figure 1: (Left) Existing methods for handling spatial structures in UDA for wearable MTS data. (Right) Examples of diverse inter-domain sensor relationships observed across scenarios.

*ther collapsed or rigidly hard-coded* as depicted in Figure 1 (Left): 1) high-level matching between source and target feature embeddings without explicit sensor alignment (Sun & Saenko, 2016; Wilson et al., 2020), or 2) strict 1:1 matching between sensors in the same location (Wang et al., 2023). Both tacitly assume fixed cross-domain sensor relations based on physical placement and rely solely on *intra-domain* graphs (e.g., using GNNs or CNNs as feature encoder (Lai et al., 2021)) to learn multi-sensor structure. This is more problematic in wearables, where intra-domain correlations are noisy and domain-specific; treating them as static across domains can entangle semantics and shift, and ultimately undercut adaptation.

The right side of Figure 1 illustrates two UDA scenarios involving three domains (people): two subjects with the same dominant hand (Green and Orange in Figure 1) and one subject with the opposite dominant hand (Yellow). The same sensor attached to the same limb across source and target may (Scenario 1) or may not (Scenario 2) produce the same distributions, depending on domains in the UDA scenario, driven by domain information such as handedness. As in this illustrative example, various dynamic inter-domain sensor relations occur, and existing approaches that assume simple or static correspondence between inter-domain sensors may fail in domain adaptation. We further substantiate the need to model inter-domain sensor relations in the real-world wearable MTS dataset through an empirical analysis in Section 3.

To fully address the sensor-wise distribution shift in wearable MTS UDA, we propose *Inter-Domain Sensor Alignment for MTS-UDA* (**IDSA**), a *plug-in* module that extends existing UDA methods (e.g., Deep CORAL (Sun & Saenko, 2016) or CLUDA (Ozyurt et al., 2023)) without architectural changes. IDSA adds a principled **inter-domain** perspective and couples it with **intra-domain** structure via domain-specific sensor embeddings and two additional losses for unified *inter-* and *intra*-domain structure in a single training routine. IDSA introduces a principled **inter-domain** component that learns a cross-sensor relation matrix from domain-specific sensor embeddings through spatial transportation loss and uses it to *transport* target channels toward the source, thereby aligning inter-domain sensor semantics. This is coupled with an **intra-domain** *channel decorrelation* regularizer that sparsifies within-domain graphs, suppressing redundant or noisy intra-domain sensor couplings so that inter-domain relations remain distinctive and informative. Crucially, our spatial transportation objective admits a clean optimal-transport interpretation: minimizing it is equivalent (up to a constant) to the discrete 1-Wasserstein objective.

We evaluate our model on five real-world multivariate time-series (MTS) wearable sensor datasets, encompassing Human Activity Recognition (HAR) under the cross-subject setting and surface electromyography (sEMG) under the cross-repetition setting. Both settings are classification tasks. In the cross-subject setting, the source and target domains are two different subjects, whereas in the cross-repetition setting, the source and target domains are two different repetitions from the same subject.

## 2 RELATED WORK

Wearable sensor technologies, often embedded in devices such as smartphones and smartwatches, have attracted significant attention owing to their accessibility and ease of use. These technologies have been widely applied to classification tasks, including activity recognition (Roggen et al., 2010) and gesture recognition (Shen et al., 2019). A notable characteristic of wearable sensor data is the variation in distributions arising from differences in subjects and wearing conditions (Wang et al., 2023). With the increasing availability of unlabeled datasets, Unsupervised Domain Adaptation (UDA) has emerged as a promising strategy for improving the performance of classification models tailored to wearable sensor data (Liu et al., 2025).

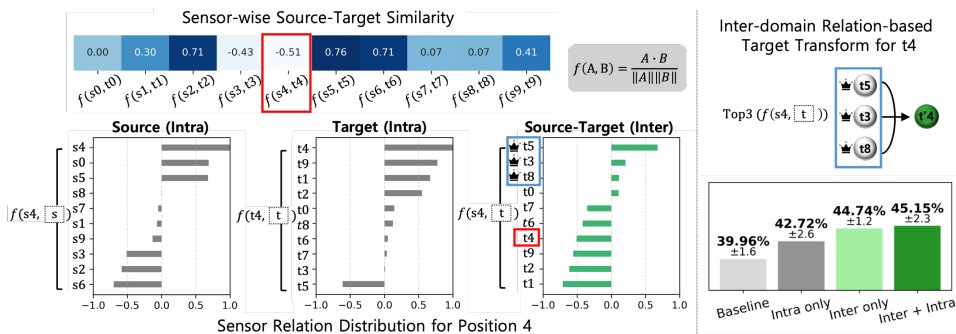

Figure 2: Illustration of the motivation analysis.

UDA seeks to learn representations or models that are robust to distributional discrepancies between the source and target domains, and has been extensively adopted in various fields (Huang et al., 2022; Ozyurt et al., 2023; Li et al., 2022). A prominent class of methods leverages adversarial training, where a discriminator is introduced to identify whether a given input representation originates from the source or target domain. (Ganin et al., 2016; Long et al., 2018). Another line of research employs statistical alignment techniques to minimize domain divergence, such as moment matching or correlation-based methods (Sun & Saenko, 2016; Redko et al., 2019). However, these UDA techniques are typically developed for univariate data. When applied to multivariate time series (MTS), a common strategy is to concatenate data from all sensors into a single representation, which refers to the left side of Figure 1-1). This practice often neglects spatial dependencies across domains, which may constrain the effectiveness of domain adaptation in wearable MTS scenarios, where spatial relations are influenced by numerous factors.

UDA for MTS data, which considers the multi-sensor structure, has been recently explored (Wang et al., 2023; Sun et al., 2024; Guo et al., 2025). For example, SEA (Wang et al., 2023) typically models intra-domain dependencies among sensors and employs a fixed sensor alignment strategy, as shown in Figure 1-2). A key underlying assumption is that corresponding sensors in the source and target domains serve equivalent semantic functions or measure similar physical phenomena. Other studies (Li et al., 2023; Cai et al., 2021) extract sparse associative structure with intra- and inter-variable attention mechanisms, further leveraging domain-invariant causal structures while also modeling domain-specific components. However, as illustrated in the right side of Figure 1, inter-domain sensor relationships may differ from a straightforward one-to-one mapping between sensors at corresponding positions. This mismatch can result in suboptimal adaptation performance due to misaligned sensor interactions across domains. A more extensive comparison with these methods is discussed in the Appendix.

## 3 EMPIRICAL ANALYSIS OF MOTIVATION

Figure 1 illustrates the importance of modeling meaningful inter-domain sensor relationships in multivariate time-series (MTS) unsupervised domain adaptation (UDA). It shows how existing methods capture spatial dependencies and highlights scenarios where their underlying assumptions may be insufficient. We conduct an empirical analysis on **real-world MTS data** to verify this motivation with two objectives: **1)** to demonstrate the existence of inter-domain sensor relationships that go beyond simple 1:1 sensor matching; and **2)** to reveal the limitations of utilizing intra-domain sensor relationships naively to address spatial distribution shift in wearable MTS data.

**Setup** Two different domains (subjects) from the Opportunity HAR dataset (Roggen et al., 2010) are selected, extracting the same 10 sensors randomly for both domains. First, we compute the cosine similarity using a function $f$ between the sensor signals, including sensors across domains in the same position, to analyze relationships from both intra- and inter-domain perspectives. Next, we design a simple target transformation based on the observed inter-domain sensor relations: for each target sensor position, we replace its signal with a weighted sum of the three target sensors that exhibit the highest similarity with the corresponding source sensor in the same position. We denote

the original target data as $\mathcal{T}$, the transformed version as $\mathcal{T}_{\text{transform}}$, and the source domain as $\mathcal{S}$. We define a UDA model $\Phi$ utilizing a Graph Neural Network structure (Kipf & Welling, 2017) and Deep CORAL loss (Sun & Saenko, 2016). Depending on the specified adjacency matrix, the model either excludes intra-domain sensor relations ($\Phi_{\text{base}}$) or incorporates intra-domain sensor relations ($\Phi_{\text{intra}}$).

**Results** Figure 2 summarizes the findings. The top left panel is a cosine similarity heatmap for position-matched sensors across source and target domains. Similarities vary widely, indicating that fixed physical alignment does not ensure functional consistency across domains. We then focus on Sensor 4, which showed the lowest inter-domain similarity among corresponding pairs. The three plots in the bottom left depict similarity distributions for Sensor 4 from the source and target intra-domain perspectives and the source–target inter-domain perspective. The intra-domain structural patterns differ notably between the source and target domains. Regarding inter-domain relationships, interestingly, sensors 5, 3, and 8 (which are not position-matched sensors) show greater similarity than the position-matched sensor (e.g., sensor 4). These findings underscore the importance of explicitly accounting for inter-domain sensor relationships beyond the intra-domain relations.

Building on this, we test whether incorporating sensor relations improves UDA performance. The bottom right panel reports accuracy (mean $\pm$ s.d., $n{=}5$ runs) under four settings: *Baseline* ($\Phi_{\text{base}}^{\mathcal{S},\mathcal{T}}$), *Intra only* ($\Phi_{\text{intra}}^{\mathcal{S},\mathcal{T}}$), *Inter only* ($\Phi_{\text{base}}^{\mathcal{S},\mathcal{T}_{\text{transform}}}$), and *Inter+Intra* ($\Phi_{\text{intra}}^{\mathcal{S},\mathcal{T}_{\text{transform}}}$). From the right side of Figure 2, applying Inter-domain relations yields performance gains consistently (Inter only > Baseline, and Inter+Intra > Intra only), showing the usefulness of leveraging them in the adaptation model. Although combining both inter- and intra-domain sensor relations yields the highest mean performance, the standard deviation remains high. In particular, the larger error bars observed with intra-domain relations suggest that they may propagate redundancy or noise when applied naively. This indicates that their integration should be handled carefully.

**Takeaway** The analysis indicates two requirements for wearable MTS UDA: (i) inter-domain sensor relations must be *discovered* in a data-dependent manner for each source and target domain pair rather than assumed by position, and (ii) intra-domain structure must be handled carefully during integration, since domain-specific correlations can either help or hinder when combined with inter-domain alignment.

## 4 PRELIMINARIES

### 4.1 PROBLEM SETUP AND FORMULATION

In a UDA setting, an input spatio-temporal dataset of wearable MTS data can be represented as $\mathcal{D}_s = \{(\mathbf{X}_s^i, y_s^i)\}_{i=1}^{M_s}$ for **labeled** source domain and $\mathcal{D}_t = \{(\mathbf{X}_t^i)\}_{i=1}^{M_t}$ for **unlabeled** target domain, where $M = M_s + M_t$ indicates the total number of data, and $\mathbf{X}^i \in \mathbb{R}^{N \times T}$ is a multivariate sensor data instance. For the dimension of each data instance, $N$ denotes the number of sensors, and $T$ represents the number of timestamps. The term sensor refers to a single unit that outputs one value per channel. Under this definition, an IMU sensor consists of six signals: three from the accelerometer and three from the gyroscope. For clarity, we will omit the notation $i$ in the following discussion. Thus, the source and target data will be denoted simply as $\mathbf{X}_s$ and $\mathbf{X}_t$, respectively.

### 4.2 SENSOR-WISE DOMAIN SHIFT

We define a new notion of domain shift in wearable MTS sensor data. This type of shift has not been sufficiently addressed in existing work, yet it represents a critical challenge.

**Definition 1** (Sensor-wise Domain Shift). *Sensor-wise domain shift in wearable multivariate time series (MTS) refers to the distributional discrepancy arising from variations in sensor configurations, such as differences in sensor placement or orientation across domains, which affect both intra- and inter-domain sensor relationships.*

As discussed in Section 3, *Sensor-wise Domain Shift* can be easily found in real-world wearable MTS data. Building on the takeaway from the motivational analysis, the following section introduces a novel plug-in method that captures inter-domain sensor relations while further refining intra-domain relations to address *Sensor-wise Domain Shift*.

Figure 3: Illustration of IDSA. IDSA is designed to minimize spatial transportation loss ($\mathcal{L}_{\text{st}}$) and channel decorrelation loss ($\mathcal{L}_{\text{dec}}$) by utilizing domain-specific sensor embeddings ($\mathbf{P}_s, \mathbf{P}_t$).

## 5 METHODOLOGY

This section introduces *Inter-Domain Sensor Alignment* (**IDSA**) for wearable MTS-UDA. Figure 3 illustrates the overall framework. IDSA is designed as a plug-in module, specifically designed to mitigate sensor-wise domain shift, which can be combined with existing domain adaptation methods and feature extractor modules. It integrates two key objectives through dedicated loss functions: (i) a spatial transportation loss, which formulates inter-domain alignment as an optimal transport problem and learns a relation matrix $\mathbf{A}_{st}$ that captures essential cross-domain sensor dependencies to further transform the target distribution, and (ii) a channel decorrelation loss, which suppresses redundant or noisy sensor relations to preserve intrinsic intra-domain spatial information. The learnable sensor embeddings for both domains are utilized in model processing with input data to couple these objectives. The embeddings encode decorrelated intra-domain features that make $\mathbf{A}_{st}$ distinctive and robust. Together, these mechanisms allow IDSA to transform the target data into representations that reflect inter-domain spatial dependencies, considering essential domain-specific characteristics.

### 5.1 INTER-DOMAIN SENSOR TRANSPORTATION

The aforementioned analyses underscore the need to account for dynamic inter-domain sensor relationships due to complex distributional discrepancies in wearable MTS data caused by sensor-wise domain shift. Inter-domain sensor alignment aims to transform the target domain to match the source domain's sensor similarity. In order to identify the inter-domain sensor relations, we define two learnable embeddings: source sensor embedding $\mathbf{P}_s \in \mathbb{R}^{N \times T}$ and target sensor embedding $\mathbf{P}_t \in \mathbb{R}^{N \times T}$, which have the identical dimension as the input of MTS data. The embeddings capture the roles of sensors specific to each domain by learning through intra-domain graph generation and feature extraction.

Moreover, these learnable embeddings can express an inter-domain sensor relation between the source and target input. The inter-domain sensor relation $\mathbf{A}_{st} \in \mathbb{R}^{N \times N}$ can be expressed as:

$$\mathbf{A}_{st} = \texttt{Norm}(\sigma(\mathbf{P}_s \mathbf{P}_t^\top)), \tag{1}$$

where $\sigma$ is an activation function, and $\texttt{Norm}$ is a normalization function. The learnable adjacency matrix $\mathbf{A}_{st}$ associates multiple sensors across different domains.

**Spatial Transportation Loss**    To effectively generate inter-domain sensor relations that mitigate sensor-wise domain shift, it is essential to account for the differences between source and target signals in a spatial perspective. Therefore, we propose a loss function to generate an inter-domain sensor relation matrix that dynamically learn how to transport the target distribution to the source distribution regarding the spatial relation across two domains. The spatial transportation loss is defined as:

$$\mathcal{L}_{st} = \sum \mathbf{A}_{st} \otimes \mathbf{C}, \quad \mathbf{C}_{ij} = \|X_t(i, \cdot) - X_s(j, \cdot)\|_2^2, \tag{2}$$

where $\otimes$ denotes the Hadamard product, and $\mathbf{C} \in \mathbb{R}^{N \times N}$ is the distance matrix, with each entry representing the squared Euclidean distance between two sensors from the source and target do-

mains, respectively. The calculation of sensor-wise distance between two $\mathbf{X}_t, \mathbf{X}_s$ can be extended to other formulas. The time complexity associated with the spatial transportation loss is $\mathcal{O}(N^2)$, and the details are provided in Appendix A.6.5. Since $N$ is typically small in wearable settings, the quadratic complexity does not pose a usability concern. We theoretically show in Section 5.4 that optimizing the derived inter-domain sensor relation yields a transport plan aligned with the Wasserstein distance.

## 5.2 Integration of Inter- and Intra-Domain Sensor Relation

We incorporate the learnable sensor embeddings $\mathbf{P}_s$ and $\mathbf{P}_t$ into the input data, thereby enriching each domain with domain-specific sensor characteristics from an intra-domain perspective. The incorporated sensor data for the source and target is expressed as:

$$\tilde{\mathbf{X}}_s = \mathbf{X}_s + \mathbf{P}_s, \qquad \tilde{\mathbf{X}}_t = \mathbf{X}_t + \mathbf{P}_t. \tag{3}$$

For the target domain, we further apply the inter-domain relation matrix $\mathbf{A}_{st}$ as a transport map to align its sensor arrangement with that of the source:

$$\hat{\mathbf{X}}_t = \mathbf{A}_{st}\,\tilde{\mathbf{X}}_t. \tag{4}$$

This adjustment aims to diminish the sensor-wise domain shift and to diminish the distribution discrepancies between the source and target domains.

To capture intra-domain spatial relationships among sensors, a graph neural network (GNN) (Kipf & Welling, 2017) is employed. A graph structure should be predefined to apply the GNN structure. In our graph, sensor channels are treated as nodes, and the connections between different sensors are represented as edges, resulting in a graph with $N$ nodes. The adjacency matrix of the intra-domain graph is denoted as $\mathbf{A}_s$ for the source graph and $\mathbf{A}_t$ for the target graph. We adopt a simple distance-aware graph generation similar to acquiring the inter-domain sensor relation, which is defined as:

$$\mathbf{A}_s = \sigma(\tilde{\mathbf{X}}_s\tilde{\mathbf{X}}_s^\top), \quad \mathbf{A}_t = \sigma(\hat{\mathbf{X}}_t\hat{\mathbf{X}}_t^\top). \tag{5}$$

The features from the aggregation process of the GNN can be represented as follows by simply adopting the GCN structure (Kipf & Welling, 2017):

$$\mathbf{Z}_s = \mathbf{A}_s\tilde{\mathbf{X}}_s\mathbf{W}_l, \quad \mathbf{Z}_t = \mathbf{A}_t\hat{\mathbf{X}}_t\mathbf{W}_l, \tag{6}$$

The extracted spatial embeddings $\mathbf{Z}_s, \mathbf{Z}_t \in \mathbb{R}^{N \times T}$ so that per-sensor, per-timestamp structure is preserved, and $\mathbf{W}_l$ is the weight parameter that does not change the dimension.

**Channel Decorrelation Loss** To further enhance the model's capability in capturing spatial relations, we propose incorporating an additional regularization loss. While the aggregation phase in the spatial layer effectively integrates information from multiple related sensors, it also obscures the independent feature characteristics of individual sensors and introduces additional noise. To preserve sensor-wise distinctiveness, we introduce a channel decorrelation regularizer computed on the spatial representations. The channel decorrelation loss can be expressed as:

$$\mathbf{D}_d \;=\; \frac{1}{T}\,\mathbf{Z}_d\mathbf{Z}_d^\top \;\in\; \mathbb{R}^{N \times N}, \qquad d \in \{s, t\}, \tag{7}$$

$$\mathcal{L}_{\text{dec}} \;=\; \sum_{d \in \{s,t\}} \big\|\mathbf{D}_d - \mathbf{I}_N\big\|_F^2, \tag{8}$$

where $\mathbf{I}_N$ is the $N \times N$ identity matrix and $\mathbf{D}_d$ denotes the feature correlation matrix constructed from the spatial outputs for each domain $d$.

The decorrelation loss affects both types of graph generation. It indirectly induces the generated intra-domain graph to have high weights on the self-loop, since that is a straightforward method to optimize the decorrelation loss function. This implies that the loss function we adopt can lead to graph sparsification, which keeps the weights of the different sensors comparatively low. Due to the sparsification, our generated graph will only contain highly related sensors since graph sparsification alleviates the redundant data and noise from the graph. These intra-domain graphs help learn each domain's sensor embedding only to contain essential domain-specific sensor characteristics, so the $\mathbf{A}_{st}$ constructed by them is more distinctive and informative.

### 5.3 TRAINING AND INFERENCE

Since IDSA is designed as a plug-in module that can be integrated with existing domain adaptation methods, we also include a base domain adaptation loss $\mathcal{L}_{da}$, for which we adopt Deep CORAL (Sun & Saenko, 2016) and CLUDA (Ozyurt et al., 2023) for the experiment. $\mathcal{L}_{da}$ additionally includes the supervised classification loss on the source domain. Besides the base domain adaptation loss, $\mathcal{L}_{st}$ and $\mathcal{L}_{dec}$, are integrated into our final loss function. Therefore, the final loss for IDSA is denoted as:

$$\mathcal{L} = \mathcal{L}_{da} + \lambda_1 \mathcal{L}_{st} + \lambda_2 \mathcal{L}_{dec}, \tag{9}$$

where $\lambda_1, \lambda_2$ are hyperparameters used to adjust the effect of the incorporated losses. Since the evaluation is performed on an unseen target domain, we apply the same target-side preprocessing pipeline at test time.

### 5.4 THEORETICAL ANALYSIS ON THE SPATIAL TRANSPORTATION LOSS

We now formalize the spatial transportation procedure in our framework by showing that it naturally aligns with an optimal transport (OT) (Kolouri et al., 2017) perspective.

**Theorem 1.** *Let wearable source MTS data $\mathbf{X}_s$ and target MTS data $\mathbf{X}_t$ be represented as sub-elements in the $\mathcal{X}$ and $\mathcal{Y}$, respectively. Optimizing $\mathcal{L}_{st}$ has the same form as solving a discrete 1-Wasserstein problem. In particular, if $\mathbf{A}_{st}$ achieves the global minimum of $\mathcal{L}_{st}$, it provides an optimal transport plan (up to constant scaling), so $\min \mathcal{L}_{st} = N \cdot W_1(\mathbf{X}_t, \mathbf{X}_s)$, where $W_1(\mathbf{X}_t, \mathbf{X}_s)$ is the discrete 1-Wasserstein distance.*

The proof of Theorem 1 is provided in the Appendix. Theorem 1 implies that, *if* we train to minimize the spatial transportation loss function, then $\mathbf{A}_{st}$ effectively solves a linear assignment problem for the sensor-level alignment. This result provides a principled justification for IDSA's effectiveness in mitigating sensor-wise domain shifts.

Furthermore, we emphasize that our OT formulation differs slightly from conventional OT approaches for domain adaptation (Kerdoncuff et al., 2020b; Courty et al., 2016b; Aritake & Hino, 2022b). While traditional methods compute an OT plan at the data-instance level, our approach operates at the sensor level, explicitly addressing sensor-wise domain shifts. A more detailed discussion of conventional OT methods is provided in the Appendix.

Another theorem is about the synergy when applying both loss functions we proposed. By optimizing the loss function, the target domain representation of IDSA satisfies the source-wise information bottleneck.

**Theorem 2.** *Let the source-wise information bottleneck be defined as:*

$$IB_s = I(\mathbf{Z}_t, \mathcal{X}_s) - \beta I(Tar, \mathbf{Z}_t), \tag{10}$$

*where $\mathcal{X}_s$ indicates the source distribution and $Tar$ denotes target domain-specific information. In particular, if IDSA is optimized through loss functions, the transformed target representation satisfies the source-wise information bottleneck.*

$$\min(\mathcal{L}_{st} + \mathcal{L}_{dec}) \Rightarrow \max IB_s. \tag{11}$$

This theorem formalizes the joint application of both loss functions to achieve a more effective transformation. By considering the complementary roles of these losses, the theorem provides a theoretical justification for their combined use in guiding the learning process. A detailed proof and derivation are provided in the Appendix.

### 5.5 EMPIRICAL ANALYSIS OF THEOREM 1

Building on our theoretical result in Theorem 1, we assess whether the learned inter-domain sensor relation $\mathbf{A}_{st}$ aligns with the ground-truth optimal transport (OT) plan in practice. Specifically, we investigate how closely $\mathbf{A}_{st}$ (trained only under $\mathcal{L}_{st}$) approximates the exact OT solution on a real sensor dataset.

**Setup** We employ the SD-gesture sEMG dataset (Lee et al., 2023), containing 8 sensor channels. To reduce each domain to a representative vector, we compute the mean value of every sensor channel. From these means, we form a distance matrix $\|\mathbf{X}_t^i - \mathbf{X}_s^j\|_2$ for every sensor pair $(i, j)$ and solve a linear assignment problem to obtain a ground-truth OT plan $\pi^*$. Meanwhile, we train $\mathbf{A}_{st}$ end-to-end via $\mathcal{L}_{st}$ alone, as per our method.

**Results** Figure 4 shows that, during training, the mean squared error between $\mathbf{A}_{st}$ and the ground-truth OT plan $\pi^*$ decreases and converges. The y-axis is the mean squared error (MSE) between our learned $\mathbf{A}_{st}$ and the ground-truth OT plan. Empirically, the figure confirms that the learned alignment matrix $\mathbf{A}_{st}$ approximates a transport plan, consistent with Theorem 1.

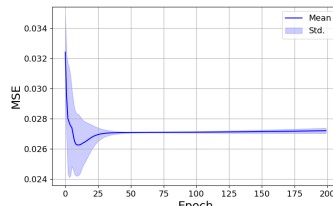

Figure 4: Convergence toward the optimal transport (OT) plan.

## 6 EXPERIMENT

### 6.1 DATASETS AND BASELINES

We conduct extensive evaluations to verify how IDSA performs and compare it with state-of-the-art methods on public wearable MTS datasets, categorized into two groups: (1) human activity recognition (HAR) datasets for cross-subject adaptation and (2) surface electromyography (sEMG) datasets for cross-repetition adaptation. We compare IDSA with ten UDA baselines: Deep Coral (Sun & Saenko, 2016), CDAN (Long et al., 2018), CoDATS (Wilson et al., 2020), AdvSKM (Liu & Xue, 2021), CLUDA (Ozyurt et al., 2023), SEA (Wang et al., 2023), RAINCOAT (He et al., 2023), ACON (Liu et al., 2024), SASA (Cai et al., 2021), and UniMTS (Zhang et al., 2024). A detailed description of the datasets and baselines is provided in the Appendix. Additional experiments, including hyperparameter sensitivity, an ablation study, and visualization, are detailed in Appendix A.4.

### 6.2 PERFORMANCE COMPARISON

**Cross-subject** The performance for cross-subject settings on two HAR datasets, the Opportunity HAR and WISDM datasets, is presented in Table 1, where the domain is defined as each subject. For the Opportunity HAR dataset, We report results on all 12 scenarios. For the WISDM dataset, we follow prior works (Ozyurt et al., 2023; Wilson et al., 2020) by evaluating on 10 selected scenarios, where the specific random scenario pairs are taken from the CLUDA (Ozyurt et al., 2023) implementation. As shown in the table, IDSA outperforms the other baselines because of reflecting inter-domain sensor alignment. Table 1 indicates that not considering inter-domain sensor alignment is insecure and may underperform in specific scenarios. For example, CoDATS (Wilson et al., 2020) severely underperforms in scenarios $19 \rightarrow 2$ on the WISDM dataset and CLUDA (Ozyurt et al., 2023) in $3 \rightarrow 1$ on the Opportunity HAR dataset. On the other hand, IDSA, which incorporates the spatial transportation loss to alleviate the sensor-wise domain shift, maintains robust performance in most scenarios. IDSA closes the gap between the two distinct distributions by estimating the Wasserstein distance between the distributions, which leads to state-of-the-art performance when comparing the average performance across all scenarios. UniMTS (Zhang et al., 2024) shows strongest performance in WISDM benefiting from an additional textual modality during pretraining, unlike ours and other baselines. On Opportunity dataset composed of heterogenous sensor set, UniMTS's relatively low performance may because its constrained IMU-only input, as it is architecturally tailored to IMU sensor-channel structure, while others use the full set of available sensors. We therefore further analyze and report an additional controlled comparison between UniMTS and our method under an IMU-only setting in Appendix.

**Cross-Repetition** Model performance on the three sEMG datasets for cross-repetition adaptation is presented in Table 2, where each domain corresponds to a specific repetition. For evaluation, we designate a sampled repetition from a subject as the source domain and the remaining repetition data from the same subject as the target domain. Across all three datasets, our model consistently outperforms existing UDA baselines, demonstrating its robustness and effectiveness in hand gesture recognition tasks. For the Nina-53 dataset, which exhibits a large gap between accuracy and F1-score due to severe label imbalance, we observe a notable improvement in F1-score when our model is added to CLUDA. In the sEMG datasets, most domain discrepancies arise due to varia-

Table 1: Overall performance comparison on cross-subject scenarios. Each scenario corresponds to a source → target scenario. The **boldfaced** score denotes the best result, and the underlined score represents the second-best result. Parentheses denote relative gains/losses over base models.

| S → T | Metric (%) | Deep Coral | CDAN | CoDATS | AdvSKM | CLUDA | SEA | RAINCOAT | ACON | SASA | UniMTS | IDSA + Deep Coral | IDSA + CLUDA |
|---|---|---|---|---|---|---|---|---|---|---|---|---|---|
| | | | | | | **Opportunity HAR** | | | | | | | |
| 1 → 2 | Acc. | 81.56 | 81.56 | 80.73 | 83.52 | 85.20 | 80.93 | 82.89 | 78.98 | 81.09 | 61.73 | **86.82** (5.26) | 86.03 (0.83) |
| | F1 | 85.07 | 84.74 | 84.10 | 86.66 | 87.96 | 82.11 | 83.56 | 83.06 | 84.65 | 64.89 | **88.69** (+3.62) | 88.09 (+0.13) |
| 1 → 3 | Acc. | 79.50 | 78.57 | 78.57 | 80.74 | 86.02 | 75.31 | 77.23 | 75.31 | 76.87 | 74.54 | 81.06 (+1.56) | **86.96** (+0.94) |
| | F1 | 62.11 | 60.52 | 60.59 | 63.26 | 87.71 | 55.11 | 58.02 | 55.12 | 51.58 | 75.86 | 63.41 (+1.30) | **88.11** (+0.40) |
| 1 → 4 | Acc. | 82.31 | 76.41 | **84.72** | 83.91 | 83.65 | 74.43 | 75.39 | 78.13 | 82.62 | 69.17 | 82.57 (+0.26) | 84.45 (+0.80) |
| | F1 | 85.78 | 59.42 | 87.19 | 86.40 | **88.91** | 82.52 | 58.99 | 82.62 | 84.71 | 74.94 | 77.11 (-8.67) | 88.89 (-0.02) |
| 2 → 1 | Acc. | 84.53 | 82.52 | 85.67 | 82.81 | 85.10 | 85.62 | 81.64 | 87.50 | 77.22 | 81.37 | **89.30** (+4.87) | 85.96 (+0.86) |
| | F1 | 87.18 | 62.76 | 87.88 | 82.23 | 87.93 | 88.75 | 65.09 | 88.75 | 78.45 | 83.12 | **90.95** (+3.77) | 88.38 (+0.45) |
| 2 → 3 | Acc. | 81.67 | 75.16 | 81.99 | 80.75 | 77.95 | 75.94 | 77.73 | 77.50 | 75.86 | 67.08 | 81.37 (-0.30) | **84.78** (+6.83) |
| | F1 | 82.74 | 58.66 | 83.42 | 77.11 | 81.12 | 58.47 | 58.45 | 60.29 | 77.38 | 42.96 | 79.42 (-3.32) | **87.15** (+6.03) |
| 2 → 4 | Acc. | 79.09 | 82.84 | 79.36 | 73.73 | **84.45** | 76.70 | 71.48 | 70.17 | 82.57 | 58.71 | 74.27 (-4.82) | 81.23 (-3.32) |
| | F1 | 82.70 | 76.12 | 84.48 | 82.68 | **89.32** | 53.70 | 75.45 | 75.00 | 74.44 | 59.43 | 80.63 (-2.07) | 87.45 (-1.87) |
| 3 → 1 | Acc. | 81.66 | 69.91 | 83.67 | 77.08 | 63.32 | 75.31 | 70.49 | **85.54** | 84.43 | 78.75 | 83.38 (+1.72) | 79.08 (15.76) |
| | F1 | 62.35 | 41.64 | **84.74** | 58.85 | 51.10 | 46.60 | 63.83 | 53.49 | 56.20 | 55.41 | 63.32 (+0.97) | 62.42 (+11.32) |
| 3 → 2 | Acc. | 66.20 | 53.91 | 62.57 | 64.25 | 76.54 | 67.61 | 78.52 | 70.17 | 67.39 | 61.73 | **81.01** (+14.81) | 80.45 (+3.91) |
| | F1 | 49.46 | 33.77 | 50.44 | 49.64 | 58.99 | 44.17 | 60.28 | 48.56 | 50.44 | 49.48 | **62.23** (+12.77) | 61.62 (+2.63) |
| 3 → 4 | Acc. | 75.60 | 64.88 | 83.11 | 78.02 | **87.67** | 66.48 | 58.20 | 66.48 | 80.35 | 58.98 | 80.43 (+4.83) | 82.84 (-4.83) |
| | F1 | 84.06 | 72.20 | 87.32 | 83.30 | **91.39** | 72.89 | 64.00 | 72.89 | 72.31 | 48.10 | 86.67 (+12.77) | 88.09 (+2.63) |
| 4 → 1 | Acc. | 81.38 | 76.79 | 87.97 | 86.24 | 86.25 | 79.69 | 87.89 | 83.13 | 88.55 | 47.85 | 83.38 (+2.00) | **93.69** (+7.44) |
| | F1 | 83.55 | 71.88 | 88.99 | 80.97 | 88.11 | 75.26 | 88.32 | 81.91 | 83.11 | 40.92 | 83.98 (+0.43) | **94.71** (+6.60) |
| 4 → 2 | Acc. | 74.58 | 68.43 | **86.31** | 73.74 | 85.75 | 76.14 | 75.00 | 77.27 | 84.04 | 46.09 | 76.82 (+2.24) | 82.12 (-3.64) |
| | F1 | 57.42 | 46.72 | **88.64** | 56.43 | 87.91 | 78.29 | 61.56 | 78.19 | 63.28 | 42.66 | 76.62 (+19.20) | 84.66 (-3.25) |
| 4 → 3 | Acc. | 71.12 | 72.05 | 79.19 | 75.15 | 84.47 | 82.81 | 69.14 | 80.94 | 85.14 | 64.91 | 76.71 (+5.59) | **88.51** (+4.04) |
| | F1 | 64.11 | 47.86 | 60.99 | 61.49 | 85.22 | 81.40 | 72.19 | 79.94 | 64.37 | 49.05 | 69.83 (+5.72) | **89.39** (+4.17) |
| Avg. | Acc. | 78.27 | 73.59 | 81.16 | 78.33 | 82.20 | 76.41 | 76.37 | 77.03 | 80.51 | 63.55 | 81.44 (+3.17) | **84.68** (+2.48) |
| | F1 | 73.88 | 59.69 | 79.07 | 72.42 | 82.14 | 68.27 | 67.48 | 71.65 | 70.08 | 57.24 | 76.91 (+3.03) | **84.08** (+1.94) |
| | | | | | | **WISDM** | | | | | | | |
| 2 → 28 | Acc. | 82.27 | 71.11 | 80.92 | 80.90 | 84.96 | 80.00 | 83.15 | 75.60 | 84.94 | 77.77 | **86.67** (+4.40) | 86.67 (+1.71) |
| | F1 | 69.64 | 44.96 | 70.93 | 54.51 | 80.48 | 69.35 | 76.62 | 74.07 | 71.81 | 73.36 | 78.33 (+8.69) | **83.47** (+2.99) |
| 7 → 2 | Acc. | 62.40 | 75.61 | 61.06 | 61.06 | 78.05 | 68.29 | 78.05 | 83.33 | 71.25 | **90.24** | 68.30 (+5.90) | 82.93 (+4.88) |
| | F1 | 53.61 | 51.79 | 48.04 | 41.95 | 59.15 | 53.61 | 56.68 | 60.25 | 49.98 | **81.11** | 55.20 (+1.59) | 73.22 (+14.07) |
| 7 → 26 | Acc. | 75.61 | 73.17 | 78.05 | 78.05 | 63.41 | 73.17 | **81.25** | 81.08 | 58.54 | | 78.05 (+2.44) | 73.17 (+0.00) |
| | F1 | 43.19 | 40.94 | 47.87 | 48.51 | 34.50 | 33.16 | 55.31 | 59.46 | 55.33 | **65.05** | 46.76 (+3.57) | 39.40 (+4.90) |
| 12 → 7 | Acc. | 59.21 | 70.83 | 72.10 | 74.20 | 79.23 | 75.00 | 79.17 | 81.81 | 67.30 | **89.58** | 83.33 (+24.13) | 81.25 (+2.05) |
| | F1 | 35.73 | 50.30 | 65.16 | 65.61 | 69.52 | 50.83 | 69.21 | 71.21 | 50.59 | **89.88** | 73.10 (+37.37) | 79.27 (+9.75) |
| 12 → 19 | Acc. | 43.33 | 46.97 | 63.38 | 63.98 | 69.44 | 54.55 | 46.97 | 50.00 | 73.58 | **89.39** | 54.54 (+11.21) | 80.30 (+10.86) |
| | F1 | 23.16 | 23.04 | 46.02 | 46.51 | 60.70 | 37.56 | 31.82 | 39.22 | 67.27 | **89.83** | 40.52 (+17.36) | 74.93 (+14.23) |
| 18 → 20 | Acc. | 38.03 | 70.73 | 63.45 | 39.03 | 78.03 | 80.49 | 76.83 | 77.35 | 63.88 | 53.66 | 68.29 (+30.26) | **82.93** (+4.90) |
| | F1 | 29.17 | 45.83 | 39.17 | 40.63 | 68.29 | 70.35 | 64.65 | 41.78 | 45.34 | 57.36 | 43.89 (+14.72) | **78.58** (+10.29) |
| 19 → 2 | Acc. | 47.35 | 34.68 | 39.58 | 43.45 | 56.01 | 41.46 | 63.41 | 81.81 | 58.59 | **85.37** | 63.41 (+16.06) | 65.85 (+9.84) |
| | F1 | 46.21 | 17.36 | 34.91 | 42.43 | 43.98 | 52.31 | 63.48 | 57.85 | 45.32 | **74.86** | 62.34 (+16.13) | 47.46 (+3.48) |
| 28 → 20 | Acc. | 73.17 | 75.61 | 74.10 | 73.17 | 80.49 | 86.34 | 85.37 | 82.22 | 94.76 | 71.17 | **97.56** (+24.39) | 87.80 (+7.31) |
| | F1 | 63.33 | 54.18 | 53.98 | 68.25 | 72.57 | 67.17 | 81.14 | 63.89 | 84.67 | 61.01 | **97.76** (+34.43) | 85.37 (+12.80) |
| 26 → 2 | Acc. | 73.77 | 61.48 | 72.70 | 62.01 | 74.08 | 53.66 | 67.07 | 80.48 | 76.72 | 70.73 | 80.50 (+6.80) | **90.24** (+3.94) |
| | F1 | 61.78 | 40.37 | 59.76 | 46.30 | 70.05 | 37.88 | 52.48 | 63.35 | 69.46 | **91.53** | 82.37 (+20.57) | 73.89 (+3.79) |
| 28 → 2 | Acc. | 64.92 | 58.03 | 71.70 | 70.69 | 74.08 | 65.85 | 87.80 | 82.22 | 67.28 | 70.73 | 75.61 (+10.71) | **90.24** (+16.14) |
| | F1 | 49.50 | 39.98 | 49.21 | 48.35 | 71.06 | 59.13 | 72.87 | 62.88 | 51.82 | 65.56 | 52.16 (+2.66) | **81.11** (+10.11) |
| Avg. | Acc. | 62.00 | 63.82 | 67.70 | 64.65 | 75.98 | 66.56 | 74.10 | 77.61 | 73.94 | 78.36 | 75.61 (+13.61) | **82.14** (+6.16) |
| | F1 | 47.53 | 40.88 | 51.51 | 50.31 | 63.03 | 53.14 | 62.43 | 53.40 | 59.16 | **74.96** | 63.24 (+15.71) | 71.67 (+8.64) |

Table 2: Performance comparison for cross-repetition adaptation on three sEMG datasets. Each cell reports the mean ± standard deviation across runs. The **boldfaced** score denotes the best, and underlined represents the second-best result. Parentheses denote gains of IDSA over its base model.

| Models | Nina5-18 | | Nina5-53 | | SD-gesture | |
|---|---|---|---|---|---|---|
| | Accuracy (%) | F1-score (%) | Accuracy (%) | F1-score (%) | Accuracy (%) | F1-score (%) |
| Deep Coral | 83.47 ±2.1 | 60.72 ±2.8 | 75.68 ±2.5 | 40.00 ±4.2 | 73.02 ±2.1 | 72.32 ±2.6 |
| CDAN | 83.48 ±1.9 | 61.11 ±2.2 | 75.82 ±3.1 | 39.75 ±5.7 | 72.84 ±2.5 | 71.87 ±2.1 |
| CoDATS | 81.74 ±1.6 | 57.43 ±3.0 | 67.58 ±2.0 | 15.74 ±3.3 | 75.48 ±2.9 | 74.90 ±2.3 |
| AdvSKM | 83.52 ±1.7 | 61.30 ±2.9 | 76.22 ±1.8 | 40.38 ±3.4 | 75.28 ±3.0 | 74.46 ±2.6 |
| CLUDA | 80.64 ±2.2 | 53.77 ±2.1 | 65.56 ±2.7 | 10.44 ±2.9 | 72.49 ±2.2 | 71.36 ±2.7 |
| SEA | 82.73 ±2.5 | 58.02 ±3.4 | 75.53 ±3.0 | 40.36 ±3.3 | 77.41 ±1.3 | 76.48 ±2.6 |
| RAINCOAT | 80.82 ±2.0 | 55.73 ±2.3 | 75.14 ±2.5 | 39.88 ±4.7 | 74.14 ±2.9 | 73.44 ±3.4 |
| ACON | 82.07 ±1.5 | 58.40 ±3.6 | 76.31 ±1.8 | 40.72 ±5.3 | 76.96 ±1.7 | 75.51 ±2.7 |
| IDSA + Deep Coral | **85.53** ±2.1 (+2.06) | **63.32** ±3.3 (+2.60) | **78.80** ±1.2 (+3.12) | **43.77** ±4.9 (+3.77) | **80.15** ±2.1 (+7.13) | **78.58** ±3.3 (+6.26) |
| IDSA + CLUDA | 81.88 ±1.7 (+1.24) | 54.42 ±3.2 (+0.65) | 70.35 ±2.4 (+4.79) | 34.50 ±3.3 (+24.06) | 75.68 ±2.3 (+3.19) | 74.15 ±2.6 (+2.79) |

tions in sensor attachment locations across repetitions within the same subject. By incorporating inter-domain sensor alignment, these sensor-wise domain shifts can be addressed more effectively, mitigating complex distribution discrepancies more successfully than existing UDA methods.

## 6.3 Analysis of Sensor-wise Domain Shift

**Sensor Permutation Experiment** We conduct a sensor-wise permutation experiment to verify how well the model responds to the drastic sensor-wise domain shift setting, similar to the right side of Figure 1. We randomly select two sensors in the target domain and swap their data to create a synthetic setting with severe sensor-wise domain shift.

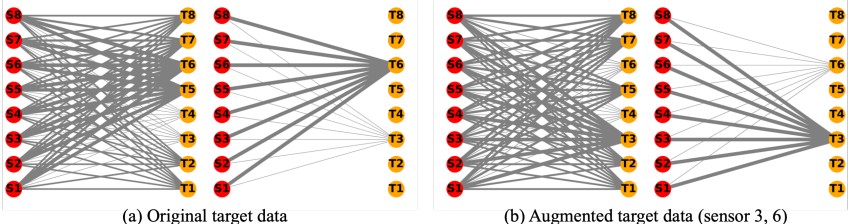

(a) Original target data        (b) Augmented target data (sensor 3, 6)

Figure 6: Transportation map visualization in a bipartite graph across source-target sensors.

Figure 5 shows the percentage of retained performance, defined as the ratio between the performance on the permuted target domain and that on the original (non-permuted) target domain, for the SD-gesture dataset. IDSA (with Deep Coral as the base), along with two baseline models (selected as the second-best in the cross-subject and cross-repetition settings, respectively) across five random sensor permutation cases, are compared. The results highlight that IDSA consistently retains the highest performance preservation rate, demonstrating its robustness to sensor permutation, an extreme sensor-wise domain shift. The actual values of the performance preservation rate are provided in the Appendix.

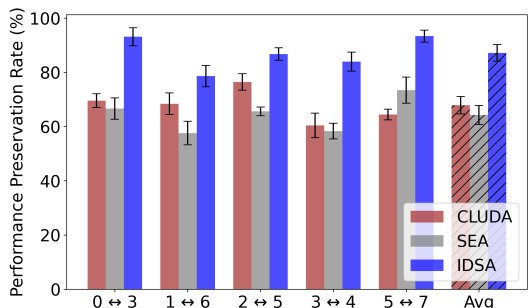

Figure 5: Performance preservation rate under sensor-wise permutations (e.g., $0 \leftrightarrow 3$ swaps the signals of sensors 0 and 3).

**Spatial Transportation Visualization** Figure 6 illustrates the learned transport map $\mathbf{A}_{\mathrm{st}}$ on the SD-gesture dataset in the cross-repetition scenario as a bipartite graph between source and target sensor channels. Figure 6 (a) shows the graph generated with the original data, and Figure 6 (b) depicts the graph with sensor-wise permutation (between sensor 3 and sensor 6) applied to the target data. For both cases, two plots are provided: one displaying the entire graph and the other highlighting only the edges corresponding to the switched target sensors. The edge thickness represents the edge weights, showing that each sensor in the target domain is transformed using a different weighted combination of the source-domain sensors. In other words, the mapping is not one-to-one; each target sensor is reconstructed from a distinct mixture of source sensors rather than being directly matched to a single source sensor. This further underscores the importance of accounting for sensor-wise domain shifts, in contrast to existing approaches that primarily assume one-to-one domain alignment or focus only on intra-domain alignment. In the sensor-wise misalignment point of view, Figure 6 (a), target sensor 6 exhibits stronger connections with source sensors, while sensor 3 shows weaker connections. In contrast, Figure 6 (b) demonstrates the opposite tendency for the swapped sensors, with target sensor 3 gaining higher relations and sensor 6 showing reduced connections. The result highlights that the IDSA can adaptively capture inter-domain sensor relationships in response to possible changes in the target domain.

## 7 CONCLUSION

We propose a plug-in module for UDA of wearable MTS sensor data, IDSA, to especially mitigate sensor-wise domain shift by exploiting the inter-domain sensor alignment. We offer theoretical insights that optimizing the proposed spatial transportation loss function provides a way to acquire the Wasserstein distance, effectively transporting the target distribution to the source distribution in spatial perspective. Extensive experiments across five benchmarks demonstrate that IDSA consistently improves over baselines, achieving substantial gains under sensor-wise shifts. These results highlight the importance of explicitly addressing sensor-wise domain shift and suggest that IDSA offers a general and effective framework for robust adaptation in the wearable domain.

## REPRODUCIBILITY STATEMENT

To ensure the reproducibility of our work, we provide our source code as supplementary material. Our theoretical analysis is presented in Section 5.4, with detailed proofs for all theorems and supporting lemmas provided in Appendix A.2. All experimental settings are described in Section 6.1 and Appendix A.5, which includes dataset statistics, the client subgraph clustering methodology, baseline model details, and a complete list of hyperparameters. We provide our implementation in the following URL: `https://anonymous.4open.science/r/IDSA_ICLR2026-ED36/`.

## ETHICS STATEMENT

This work is grounded in sensor-based multivariate time series (MTS) data, which inherently captures high-dimensional, temporally structured signals. As such, the proposed framework is not limited to a specific domain or dataset, but can be readily extended to other settings involving multi-dimensional time series. This generality enables its application across a wide range of domains, such as healthcare and industrial monitoring, where sensor networks are used to collect complex, temporally evolving data. By supporting flexible integration with diverse MTS datasets, our approach has the potential to contribute broadly to real-world deployments requiring robust, adaptable, and interpretable models.

While we acknowledge the broader concerns associated with machine learning technologies (e.g., surveillance, profiling, or disinformation), our method poses relatively low risk in these areas. Specifically, the approach operates on sensor data that does not identify personal information. Furthermore, it does not involve generating or manipulating content, nor does it make autonomous decisions that directly affect individuals. As such, while the responsible use of any AI system remains essential, the likelihood of our method being misused for harmful societal purposes is comparatively limited.

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

## APPENDIX OVERVIEW

This appendix is organized as follows. All experiments were conducted on an NVIDIA RTX A6000 GPU.

# A APPENDIX

## A.1 FEATURE EXTRACTOR OF IDSA

Following prior studies (Ozyurt et al., 2023), we adopt a TCN-based structure for the feature extractor. Therefore, after applying IDSA, the TCN-based feature extractor is applied to extract meaningful representations. The detailed structure of TCN is as follows: for the Opportunity HAR dataset, the feature extractor consists of three layers of TCN structure with a hidden dimension of 64, and five layers with a 32-dimensional hidden dimension for the WISDM dataset.

## A.2 PROOF OF THEOREMS

### A.2.1 DEFINITION OF DISCRETE 1-WASSERSTEIN DISTANCE

First, we introduce the definition of the discrete version of the Wasserstein distance as follows:

**Definition 2** (Discrete 1-Wasserstein distance (Kolouri et al., 2017)). *Suppose that when $\mathcal{X}$ and $\mathcal{Y}$ are discrete sets, each having total mass 1 (i.e., each point has mass $1/n$ and $1/m$ respectively). $n$ and $m$ denote the number of data points of $\mathcal{X}$ and $\mathcal{Y}$. Then, using the Euclidean distance, the 1-Wasserstein (or Earth Mover's) distance for the discrete distribution is defined as:*

$$W_1(\mathcal{X}, \mathcal{Y}) = \min\left(\sum_i^n \sum_j^m \pi_{i,j}\|\mathcal{X}_i - \mathcal{Y}_j\|_2\right) \ s.t. \ \pi \in \Pi, \tag{12}$$

*where $\Pi$ is $n \times m$ nonnegative coupling matrices whose row sums and column sums are equal to $1/n$ and $1/m$, respectively.*

### A.2.2 PROOF OF THEOREM 1

We restate Theorem 1 as follows:

**Theorem 3.** *Under the above notation, let source MTS data $\mathbf{X}_s$ and target MTS data $\mathbf{X}_t$ be represented as sub-elements in the $\mathcal{X}$ and $\mathcal{Y}$, respectively. Optimizing $\mathcal{L}_{st}$ has the same form as solving a discrete 1-Wasserstein problem. In particular, if $\mathbf{A}_{st}$ achieves the global minimum of $\mathcal{L}_{st}$, it provides an optimal transport plan (up to constant scaling), so $\min \mathcal{L}_{st} = N \cdot W_1(\mathbf{X}_t, \mathbf{X}_s)$.*

*Proof.* We begin the proof by assuming that the `Norm` function in equation 1 yields a doubly stochastic matrix, which can be ensured with a slight variation through the Sinkhorn–Knopp algorithm Knight (2008). From equation 2, we introduce the $\mathcal{L}_{st}$ here as follows, where $(\mathbf{A}_{st})_{i,j}$ represents the element in the $i$-th row and $j$-th column of the matrix $\mathbf{A}_{st}$:

$$\mathcal{L}_{st} = \sum_i^N \sum_j^N (\mathbf{A}_{st})_{i,j} * \text{distance}((\mathbf{X}_t)_i, (\mathbf{X}_s)_j). \tag{13}$$

Optimizing $\mathcal{L}_{st}$ leads to the following:

$$\min \mathcal{L}_{st} = \min \sum_i^N \sum_j^N (\mathbf{A}_{st})_{i,j}\|(\mathbf{X}_t)_i - (\mathbf{X}_s)_j\|_2.$$

Because $\mathbf{A}_{st}$ is doubly stochastic, each row/column sums to 1. Consequently, $\sum_{i,j}(\mathbf{A}_{st})_{i,j} = N$. Define $\widetilde{\pi} = \frac{1}{N}\mathbf{A}_{st}$. Then $\widetilde{\pi}$ has row- and column-sums $1/N$, placing it precisely in the set $\Pi$ of valid OT couplings in equation 12. Therefore,

$$\sum_{i,j}(\mathbf{A}_{st})_{i,j}\|\mathbf{X}_t^i - \mathbf{X}_s^j\|_2 = N \sum_{i,j}\widetilde{\pi}_{i,j}\|\mathbf{X}_t^i - \mathbf{X}_s^j\|_2.$$

Minimizing the left-hand side with respect to $\mathbf{A}_{st}$ is thus equivalent to minimizing the right-hand side with respect to $\widetilde{\pi} \in \Pi$. The latter is *exactly* the 1-Wasserstein cost (as in Def. fdef:Wasserstein) up to a factor $N$. Consequently:

$$\min_{\mathbf{A}_{st}} \mathcal{L}_{st} = N \min_{\widetilde{\pi} \in \Pi} \sum_{i,j}\widetilde{\pi}_{i,j}\|\mathbf{X}_t^i - \mathbf{X}_s^j\|_2 = N W_1(\mathbf{X}_t, \mathbf{X}_s).$$

Thus, the statement holds true. $\square$

### A.2.3 PROOF OF THEOREM 2

First, an assumption that the source data instance and target data instance are required. However, this assumption is reasonable because, during mini-batch training, source and target instances can be paired by their ground-truth labels, ensuring that each matched pair belongs to the same class.

We restate Theorem 2 as follows:

**Theorem 4.** *Let the source-wise information bottleneck be defined as:*

$$IB_s = I(\mathbf{Z}_t, \mathcal{X}_s) - \beta I(Tar, \mathbf{Z}_t), \tag{14}$$

*where $\mathcal{X}_s$ indicates the source distribution and $Tar$ denotes target domain-specific information. In particular, if IDSA is optimized through loss functions, the transformed target representation satisfies the source-wise information bottleneck.*

$$\min(\mathcal{L}_{st} + \mathcal{L}_{dec}) \Rightarrow \max IB_s. \tag{15}$$

*Proof.* With the assumption of identical class, the representation $\mathbf{Z}_s$ becomes similar to $\mathbf{Z}_t$ since a similar representation of identical class makes the final classifier easier to classify. Therefore, we can conclude that optimizing $\mathcal{L}_{st}$ is identical to $\mathbf{Z}_t \approx \mathbf{Z}_s$. Finally, we can express the relation of $\mathcal{L}_{st}$ as:

$$\min \mathcal{L}_{st} \approx \max I(\mathbf{Z}_t, \mathcal{X}_s), \tag{16}$$

The decorrelation loss for the target domain is related to minimizing the mutual information of the generated target representation and the target-specific domain information.

$$\min \mathcal{L}_{dec} \approx \max H(\mathbf{Z}_t), \tag{17}$$

indicating that the spatial decorrelation loss maximizes the entropy of the target-domain's representation $\mathbf{Z}_t$. If we assume that $\mathbf{Z}_t$ obeys a gaussian distribution, the entropy of $\mathbf{Z}_t$ can be transformed as:

$$
\begin{aligned}
H(\mathbf{Z}_t) &= -\int p(\mathbf{Z}_t) \log p(\mathbf{Z}_t) d\mathbf{Z}_t \\
&= -\mathbb{E}[\log \mathcal{N}(\mu_t, \boldsymbol{\Sigma}_t)] \\
&= -\mathbb{E}[\log[(2\pi)^{-D/2}|\boldsymbol{\Sigma}_t|^{-1/2}\exp(-\frac{1}{2}(\mathbf{Z}_t - \mu_t) \\
&\quad \boldsymbol{\Sigma}_t^{-1}(\mathbf{Z}_t - \mu_t))]] \\
&= \frac{D}{2}\log(2\pi) + \frac{1}{2}\log|\boldsymbol{\Sigma}_t| + \frac{1}{2}\mathbb{E}[(\mathbf{Z}_t - \mu_t) \\
&\quad \boldsymbol{\Sigma}_t^{-1}(\mathbf{Z}_t - \mu_t)] \\
&= \frac{D}{2}(1 + \log(2\pi)) + \frac{1}{2}|\boldsymbol{\Sigma}_t|,
\end{aligned}
\tag{18}
$$

where $\mu_t, \boldsymbol{\Sigma}_t$ denotes the mean and the covariance of $\mathbf{Z}_t$. $|\boldsymbol{\Sigma}_t|$ is the determinant of the covariance matrix of $\mathbf{Z}_t$. Therefore, maximizing the entropy is identical to maximizing the covariance matrix. If we assume that $\lambda_1, \lambda_2, ..., \lambda_N$ are the $N$ eigenvalues of $\boldsymbol{\Sigma}_t$, then $\sum_{i=1}^{N} \lambda_i = \text{trace}(\boldsymbol{\Sigma}_t) = N$. Finally, we have the following equation:

$$\log|\boldsymbol{\Sigma}_t| = \log \Pi_{i=1}^{N}\lambda_i = \sum_{i=1}^{N}\log\lambda_i \le N\log\frac{\sum_{i=1}^{N}\lambda_i}{N} = 0. \tag{19}$$

The inequality is due to Jensen's Inequality. The equation indicates that the upper bound of $|\boldsymbol{\Sigma}_t|$ is 1, and it is satisfied when the covariance matrix is the identity matrix. Therefore, maximizing the entropy of the target representation matches with optimizing the decorrelation loss. With the two relations, we can easily extend that optimizing both loss functions leads the target representation to satisfy the source-wise information bottleneck $IB_s$. □

### A.3 DATASETS

#### A.3.1 HAR DATASET (CROSS-SUBJECT SCENARIO)

The Opportunity HAR dataset (Roggen et al., 2010) comprises signals from 113 sensors placed at various locations on the body. It contains recordings from 4 subjects, each of which is treated as a separate domain. The dataset provides two levels of label annotation: (1) locomotion, representing low-level activities such as sitting, standing, walking, and lying; and (2) gestures, representing 17 higher-level actions. Following prior work (Wang et al., 2023), we focus on the locomotion label annotations, resulting in a 4-class classification setting. To be specific, only the sensors that are attached to the subject are regarded, resulting in a total of 113 sensors. Spatio-temporal data is constructed by adopting 128 timestamps as one data point $\mathbf{X}$, and the overlapping ratio is 50%. We conduct an experiment by setting one subject as a source dataset and another subject as a target dataset, resulting in 12 scenarios. We randomly select six scenarios from 12 to compare the performance with baselines. Each subject's recording is repeated five times. The first three repetitions of both the source and target subjects are used as the training dataset, while the remaining two repetitions of the target dataset are used for validation and testing, respectively.

The WISDM dataset (Kwapisz et al., 2011) consists of signals from a 3-axis accelerometer collected from 30 subjects. The data is recorded from both a smartphone and a smartwatch at a sampling rate of 20 Hz. The labels represent six types of human activities: walking, jogging, sitting, standing, walking upstairs, and walking downstairs. Label imbalance exists across subjects (domains). Spatio-temporal data is constructed by segmenting the signals into non-overlapping windows of 128 timestamps per data point, denoted as $\mathbf{X}$.

#### A.3.2 SEMG DATASET (CROSS-REPETITION SCENARIO)

Three real-world sEMG datasets are used to evaluate the proposed model on a hand gesture recognition task: Ninapro DB 5 (Pizzolato et al., 2017) with 18 gestures (Nina5-18), Ninapro DB 5 with 53 gestures (Nina5-53), and static and dynamic gesture (SD-gesture) (Lee et al., 2023). All three datasets are sparse sEMG datasets, each containing a small number of sEMG electrodes. In the sEMG datasets, each repetition manifests considerable variation in distribution due to spatially misaligned sensors resulting from the reattachment process between repetitions (Farina et al., 2014). For this reason, we consider each repetition within a single subject as a domain in the sEMG datasets. The Nina5-18 and the Nina5-53 datasets have 6 repetitions each, while the SD-gesture dataset has 4 repetitions.

NinaPro DB 5 is one of 10 large databases of the NinaPro dataset, which is a representative sEMG dataset frequently used for verification in this domain (Josephs et al., 2020; Ketykó et al., 2019; Côté-Allard et al., 2017). The dataset is measured with two wearable MYO armband devices, each equipped with 8 channels, resulting in a total of 16 channels. Furthermore, the NinaPro DB 5 is comprised of three hand exercise sets, which are denoted as Exercises A, B, and C. Nina5-18 utilizes only the label set of Exercise B, which contains isometric and isotonic hand configurations, as well as basic wrist movements. Nina5-53 additionally utilizes a label set of Exercises A and C, which are basic movements of the fingers and grasping and functional movements, respectively. For both Nina5-18 and Nina5-53, the 6 repetitions for each subject are divided into two parts with a ratio of 2:4. Each part is considered a distinct domain. For evaluation, the first part is utilized as the source dataset, half of the second segment is allocated to the target dataset, and the remainder is designated as the test dataset.

SD-gesture is a sEMG dataset with an 8-channel device, conducted on 9 subjects (Lee et al., 2023). The paper that introduces this dataset focuses on static and dynamic gestures, and in our paper, we refer to this dataset as SD-gesture. A dynamic gesture indicates that the sensor data for this label is acquired when the subject performs an active movement rather than a paused gesture. The SD-gesture comprises 18 gestures (14 static and 4 dynamic). The whole 4 repetitions for each subject are divided into two parts with a ratio of 2:2. The first part is designated as the source dataset. Similar to the NinaPro DB 5, half of the second part is reserved for the target domain, while the remainder is used for testing.

## A.4 BASELINES

We compared IDSA with 10 recent UDA baselines (Deep Coral (Sun & Saenko, 2016), CDAN (Long et al., 2018), CoDATS (Wilson et al., 2020), AdvSKM (Liu & Xue, 2021), CLUDA (Ozyurt et al., 2023), SEA (Wang et al., 2023), RAINCOAT (He et al., 2023), ACON (Liu et al., 2024), SASA (Cai et al., 2021), and UniMTS Zhang et al. (2024)). Each UDA model is described in the following.

- Deep Coral (Sun & Saenko, 2016) aligns the correlation statistics of the source and target distributions by processing an MLP transformation similar to our invariance learning loss.

- CDAN (Long et al., 2018) is an extended version of DANN. It introduces a conditional adversarial domain adaptation method that leverages disentangled and transferable representations by embedding adversarial learning into deep networks. They exploit classifier predictions to condition the adversarial models, employing conditioning strategies: multi-linear conditioning and entropy conditioning.

- CoDATS (Wilson et al., 2020) attempts the domain adaptation for the time-series data. They propose a weak supervision model for the target labels, which are more readily obtainable than true labels. They additionally train the domain classifier to identify if the data originated from a source or target.

- AdvSKM (Liu & Xue, 2021) is a model that attempts the statistical technique (Maximum Mean Discrepancy (MMD)) on the univariate time-series data. A hybrid spectral kernel network is proposed, in which the first kernel adjusts the MMD metric suitable for time-series data. Furthermore, they apply adversarial learning to discriminate between the representations of source and target data.

- CLUDA (Ozyurt et al., 2023) proposes a contrastive learning framework to learn representations in multivariate time-series data. They capture the relations between the source and target through a nearest-neighbor contrastive learning method. Additionally, they utilize adversarial training to enhance the performance in multivariate time-series UDA settings.

- SEA (Wang et al., 2023) introduces an unsupervised domain adaptation method for multivariate time-series sensor data. For the domain adaptation method, they introduce an endo-feature alignment loss, composed of a sensor-correlation alignment loss, a sensor-feature alignment loss, and an exo-feature alignment loss, which leads to obtaining local and global sensor-level embeddings, respectively.

- RAINCOAT (He et al., 2023) is a domain adaptation method for time series that can handle both feature and label shifts. They address feature and label shifts by considering both temporal and frequency features, aligning them across domains, and correcting for misalignments to facilitate the detection of private labels. Raincoat improves transferability by identifying label shifts in target domains.

- ACON (Liu et al., 2024) uncovers the characteristics of both temporal features and frequency features, claiming that they cannot be equally treated in transfer learning. ACON contains three key aspects: a multi-period feature learning module to enhance the discriminability of frequency features, a temporal-frequency domain mutual learning module, and a domain adversarial learning module in the temporal-frequency correlation subspace.

- SASA (Cai et al., 2021) is a time-series domain adaptation method that aligns sparse associative structures across domains. It extracts intra- and inter-variable association graphs using attention mechanisms, then matches these graphs between source and target to capture domain-invariant dependencies, while also allowing for domain-specific components. In contrast to our sensor-level transport, SASA operates under the assumption that inter-variable relationships are largely stable across domains.

- UniMTS (Zhang et al., 2024) is a pretraining and finetuning approach for motion time series data that inputs only IMU sensors. To ensure a fair and consistent evaluation, we conduct experiments on both the WISDM and Opportunity datasets. While WISDM consists exclusively of IMU sensor signals, the Opportunity dataset includes a heterogeneous combination of IMU and additional sensor modalities. For this baseline, we restrict the input to IMU signals only, reflecting the methodological constraints. On the pretraining

Table 3: Hyperparameters to reproduce our sEMG results.

| Hyperparameter | Nina5-18 | Nina5-53 | SD-gesture |
|---|---|---|---|
| lr | 0.001 | 0.001 | 0.001 |
| batch size | 64 | 128 | 32 |
| $\lambda_1$ | 1e-1 | 1e-2 | 1e-1 |
| $\lambda_2$ | 1e-0 | 1e-1 | 1e-1 |

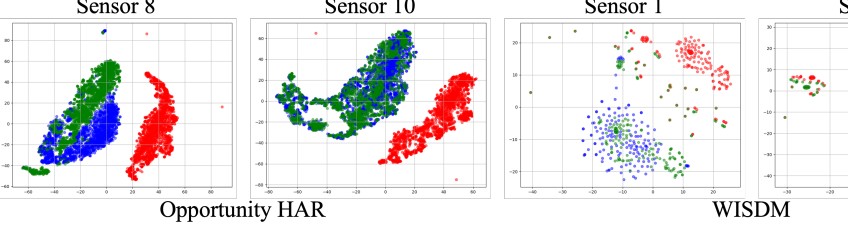

Figure 7: The t-SNE visualization of selected sensor with the source dataset, target dataset, and the transformed target representation.

stage, they apply random sensor configuration mixing augmentation to handle various sensor misalignments. The pretraining stage is trained with synthetic datasets. With the domain generalized model, they adopt a finetuning approach for domain adaptation as well. For a fair comparison, we fine-tuned UniMTS on the source training data from the pretrained model. However, they utilize the text information of the labels, which leads to extra information compared to other existing methods.

### A.5 REPRODUCIBILITY AND PERFORMANCE ANALYSIS

#### A.5.1 ANALYSIS OF HYPERPARAMETERS

To ensure the reproducibility of our model, we list the hyperparameters used in IDSA in Table 3. These parameters were selected based on the validation splits across all datasets. The search space for the learning rate includes [0.001, 0.005, 0.01, 0.05], and for batch size, we consider [128, 256, 512, 1,024, 2048]. For hyperparameters related to the loss functions ($\lambda_1$, $\lambda_2$), our search space corresponds to [0.01, 0.1, 1, 2, 10]. The embedding dimension of the Temporal layer is experimented with values ranging from 32 to 512, specifically using powers of 2 within this range.

### A.6 EXTENDED EXPERIMENTS

**Sensor-aware Visualization** To further validate the effectiveness of our method, we conduct an additional experiment demonstrating that transforming target data using the inter-domain sensor relation matrix effectively mitigates sensor-wise domain shifts. We visualize three types of single-sensor representations using t-SNE: source, target, and transformed target representations (via the learned matrix).

Figure 7 presents t-SNE (Van der Maaten & Hinton, 2008) results for sensors 8 and 10 from Opportunity HAR and sensors 1 and 2 from WISDM. The original source and target representations show a clear separation, indicating sensor-level domain shifts. After applying the transformation, the target representations align closely with the source.

These results confirm that inter-domain transformation reduces domain discrepancy by aligning sensor-wise distributions, making target data more compatible with the source-trained model. The effect is more pronounced in Opportunity HAR (113 sensors) than WISDM (3 sensors), indicating that richer sensor information enables more precise alignment and improved target performance.

Table 4: Ablation study of $\mathcal{L}_{st}$ and $\mathcal{L}_{dec}$.

| $\mathcal{L}_{st}$ | $\mathcal{L}_{dec}$ | **Nina5-18** | **Nina5-53** | **SD-gesture** |
|---|---|---|---|---|
| × | × | 83.37 | 75.63 | 73.02 |
| × | ○ | 82.67 | 75.83 | 76.92 |
| ○ | × | 84.98 | 76.89 | 79.27 |
| ○ | ○ | 85.53 | 78.80 | 80.15 |

Table 5: Performance comparison of sensor-wise permutation experiment on the SD-gesture dataset. The performance of five random sensor-wise permutation cases is presented. The result shows the actual model performance value according to the preservation rate.

| Models | $0 \leftrightarrow 3$ | $1 \leftrightarrow 6$ | $2 \leftrightarrow 5$ | $3 \leftrightarrow 4$ | $5 \leftrightarrow 7$ | **Avg. Preservation Rate** |
|---|---|---|---|---|---|---|
| CLUDA | 69.56% | 68.41% | 76.45% | 60.42% | 64.43% | 67.05% |
| SEA | 66.60% | 57.59% | 65.60% | 58.33% | 73.41% | 64.31% |
| IDSA | 93.13% | 78.62% | 86.75% | 83.94% | 93.36% | 87.56% |

### A.6.1 ABLATION STUDY

Table 4 shows the effectiveness of the newly proposed loss functions. The performances in the three datasets, along with the existence of the loss functions, are exhibited in the table. From the table, we can easily observe that the best results in all datasets are when both losses are incorporated. Furthermore, accuracy increases significantly if we compare the existence of spatial transportation loss to the values without applying every loss function. This result further strengthens our assumption that identifying inter-domain sensor relations in spatial transportation loss is essential for complicated distribution shifts in MTS sensor data. Moreover, jointly applying both loss functions yields representations that behave as an effective information bottleneck, indicating that their combined use is essential for improving domain adaptation performance.

**Analysis of Hyperparameter Sensitivity** Figure 8 illustrates the variation in accuracy across different values of $\lambda$ for the two loss functions. For both functions, performance improves up to a peak and subsequently exhibits a slight decline. Notably, the patterns across scenarios demonstrate subtle differences. These observations underscore the importance of selecting appropriate $\lambda$ values to enhance overall performance. In particular, values within the range [0,1] strike a balance between performance and stability, whereas excessively high or low values tend to introduce performance inconsistencies.

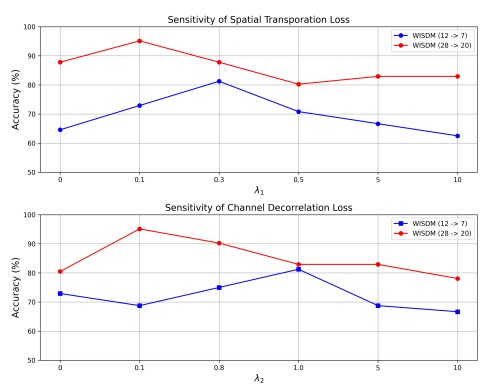

Figure 8: Hyperparameter sensitivity.

### A.6.2 VISUALIZATION OF DECORRELATION LOSS

This section visualizes the resultant decorrelation matrix $\mathbf{D}_{s,t}$ of IDSA. The decorrelation loss is designed to remove target-specific information by encouraging the embedding's correlation matrix to approximate the identity matrix. To verify this effect, we visualize the correlation matrix of the learned representations on the Nina5-53 dataset, as shown in Figure 9. The resulting matrix closely resembles the identity matrix, indicating that the learned features are effectively decorrelated. This demonstrates that the model successfully suppresses

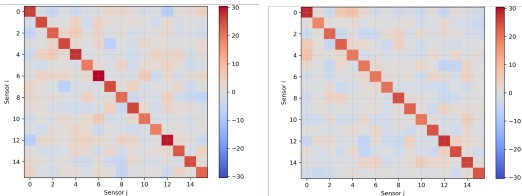

Figure 9: Intra-domain sensor correlation heatmaps on Nina5-53 dataset. (Left: Source, Right: Target)

| Scenario | UniMTS-Acc | UniMTS-F1 | Ours(IMU)-Acc | Ours(IMU)-F1 | Ours-Acc | Ours-F1 |
|---|---|---|---|---|---|---|
| $1 \to 2$ | 61.73 | 64.89 | 77.65 | 81.77 | 86.82 | 88.69 |
| $1 \to 3$ | 74.54 | 75.86 | 79.50 | 66.61 | 86.96 | 88.11 |
| $1 \to 4$ | 69.17 | 74.94 | 73.19 | 77.47 | 84.45 | 88.89 |
| $2 \to 1$ | 81.37 | 83.12 | 82.80 | 84.76 | 89.30 | 90.95 |
| $2 \to 3$ | 67.08 | 42.96 | 67.39 | 39.67 | 84.78 | 87.15 |
| $2 \to 4$ | 58.71 | 59.43 | 65.15 | 48.75 | 81.23 | 87.45 |
| $3 \to 1$ | 70.49 | 55.41 | 86.82 | 90.54 | 83.38 | 63.32 |
| $3 \to 2$ | 61.73 | 49.48 | 73.46 | 72.69 | 81.01 | 62.23 |
| $3 \to 4$ | 58.98 | 48.10 | 65.15 | 52.80 | 82.84 | 88.09 |
| $4 \to 1$ | 47.85 | 40.92 | 81.95 | 83.25 | 93.69 | 94.71 |
| $4 \to 2$ | 46.09 | 42.66 | 76.82 | 55.32 | 82.12 | 84.66 |
| $4 \to 3$ | 64.91 | 49.05 | 77.02 | 76.21 | 88.51 | 89.39 |

Table 6: Opportunity Dataset Results with UniMTS under various settings.

target-domain–specific dependencies, thereby promoting better alignment with the source domain.

### A.6.3 DETAILED RESULTS OF SENSOR-WISE DOMAIN SHIFT ANALYSIS

Table 5 presents the actual model performance values corresponding to Figure 5, which visualizes the analysis of sensor-wise permutation experiment. Our model achieves the highest performance across all cases compared to two baselines, which rank second-best on the HAR and sEMG datasets, respectively. Furthermore, the performance preservation rate of our model consistently surpasses that of the baselines by a significant margin across all cases. This result highlights the effectiveness of our approach in addressing sensor-wise domain shifts by leveraging the inter-domain sensor alignment method.

### A.6.4 PERFORMANCE COMPARISON WITH UNIMTS

Table 6 presents a fair comparison of UniMTS on the Opportunity dataset. As described in the Experiment section, UniMTS operates exclusively on IMU sensors and thus relies on a partial subset of the available sensors. In contrast, the other baselines and IDSA are designed to utilize the full sensor set. To ensure a controlled comparison, we additionally evaluate IDSA using only the IMU sensors accessible to UniMTS. From Table 6, we observe that even under this restricted sensor configuration, IDSA consistently outperforms UniMTS across most transfer settings. This result is notable, given that UniMTS benefits from supplementary textual label information, whereas IDSA does not.

### A.6.5 TIME COMPLEXITY OF THE SPATIAL TRANSPORTATION LOSS

The time complexity of $\mathcal{L}_{st}$ corresponds to $\mathcal{O}(N^2)$, as the inter-domain sensor relations and distance matrix calculation take both $\mathcal{O}(N^2)$. We observe that the number of sensors is 113 or 3 for HAR datasets and 8 or 16 for sEMG datasets, indicating that the time complexity of $\mathcal{L}_{st}$ remains computationally feasible. In cases where the number of sensors is large, we extract top-$k$ values to reduce computational overhead.

The 1-d Wasserstein distance is known to be computed using the Hungarian algorithm (Kuhn, 1955), in which the time complexity is $\mathcal{O}(N^3)$. Therefore, optimizing $\mathcal{L}_{st}$ can be stated as an efficient way of estimating the Wasserstein distance. In contrast to efficient methods for acquiring optimal transport, such as the Sinkhorn algorithm (Cuturi, 2013), IDSA is a learning-based approach. This distinction implies that IDSA eliminates the need for additional estimation or approximation iterations, thereby enhancing time efficiency.

Table 7: Comparison of domain types, variable sets, and tasks across methods.

| Method | Domain Examples | Variable Set Example | Task |
|---|---|---|---|
| SASA (Cai et al., 2021) | Regions, Age-based groups | (Blood Glucose, Glucagon, Insulin) | Timeseries UDA |
| GCA (Li et al., 2023) | Simulations, Regions | (Blood Glucose, Glucagon, Insulin) | Semi-supervised timeseries forecasting DA |
| IDSA (Ours) | Subjects, Repetitions | IMU/sEMG sensor channels (channel 1, channel 2, …, channel n) | Timeseries UDA |

Table 8: Domain-similarity statistics across four datasets.

| Dataset | Domains | # Variables | Mean Sim. (%) | Std (%) | Min–Max (%) |
|---|---|---|---|---|---|
| Opportunity | 4 subjects | 113 | 44.89 | 11.57 | 35.44–69.83 |
| WISDM | 30 subjects | 3 | 66.61 | 18.71 | 40.19–88.06 |
| Air quality | 4 cities | 11 | 81.29 | 7.91 | 67.49–91.75 |
| Boiler | 3 conditions | 20 | 94.38 | $\sim 2.7$ | 92.47–97.84 |

## A.7 EXTENDED RELATED WORK

### A.7.1 DISCUSSION WITH SIMILAR STRUCTURAL ALIGNMENT METHODS IN MTS-UDA

A growing line of research explicitly models and *aligns inter-variable structure* across domains for time-series DA. Representative examples include SASA (Cai et al., 2021), which learns sparse associative graphs using intra- and inter-variable attention and aligns them across domains, and GCA (Li et al., 2023), which formalizes causal conditional shift and extracts a domain-invariant causal skeleton while allowing domain-specific components for forecasting. These methods are compelling when variable semantics are stable across domains (e.g., clinical or environmental variables whose relationships vary slowly, such as Glucagone and Insulin), and they have demonstrated strong transfer under that assumption. Let $X = (X_1, \ldots, X_N)$ denote variables (channels), $\varphi$ a feature map, and $C$ their structural dependency (associative/causal). Structure-alignment methods typically operate under

$$P_S(y \,|\, \varphi(X)) \neq P_T(y \,|\, \varphi(X)), \qquad P_S(C \,|\, \varphi(X)) \approx P_T(C \,|\, \varphi(X)),$$

i.e., the inter-variable structure is approximately shared while label conditionals (or marginals) can shift.

Table 7 summarizes the conceptual differences between SASA, GCA, and IDSA in domain types, variable sets, and tasks. Our setting targets wearable MTS under cross-subject or cross-repetition domain shift scenarios, where each "variable" is a sensor channel whose semantics can easily change across domains due to placement, orientation, or user characteristics. In such cases, the assumption of a single shared inter-variable structure often fails. We therefore adopt the hypothesis that regards sensor-wise domain shift:

$$P_S(C \,|\, \varphi(X)) \neq P_T(C \,|\, \varphi(X)), \qquad P_S(C \,|\, \varphi(X)) \approx P_T(C \,|\, \varphi(X), \mathbf{A}_{st}),$$

where $\mathbf{A}_{st} \in \mathbb{R}^{N \times N}$ denotes a cross-domain sensor correspondence. Accordingly, IDSA learns an explicit channel-wise transport map to realign target channels to the source before reasoning about structure, and couples this with intra-domain channel decorrelation to suppress spurious couplings. To verify the conceptual difference in the empirical view, we proceeded to conduct the following experiment. The experiment examined differences in inter-variable correlation consistency across domains within the dataset based on variables.

**Inter-variable relation difference analysis.** To examine whether a single shared correlation or covariance structure is appropriate for multivariate sensor data, we compute inter-variable correlation matrices for each domain within the same dataset. This procedure is repeated independently for all four datasets, using PCA-stabilized channel statistics. The resulting domain similarity is defined as the cosine similarity between the vectorized correlation matrices. For each domain $d$ in a dataset, we compute per-channel statistics (mean and standard deviation), concatenate them to form $F_d \in \mathbb{R}^{T \times 2N}$, z-score features within domain, and apply PCA fitted on $F_d$ to retain 95% variance to stabilize dimensionality. We then derive a variable-variable correlation matrix $C_d \in \mathbb{R}^{N \times N}$ by Pearson correlation using the PCA-stabilized features. Domain similarity between two domains $i, j$ is defined as the cosine similarity of the vectorized upper-triangular (excluding diagonal) parts of

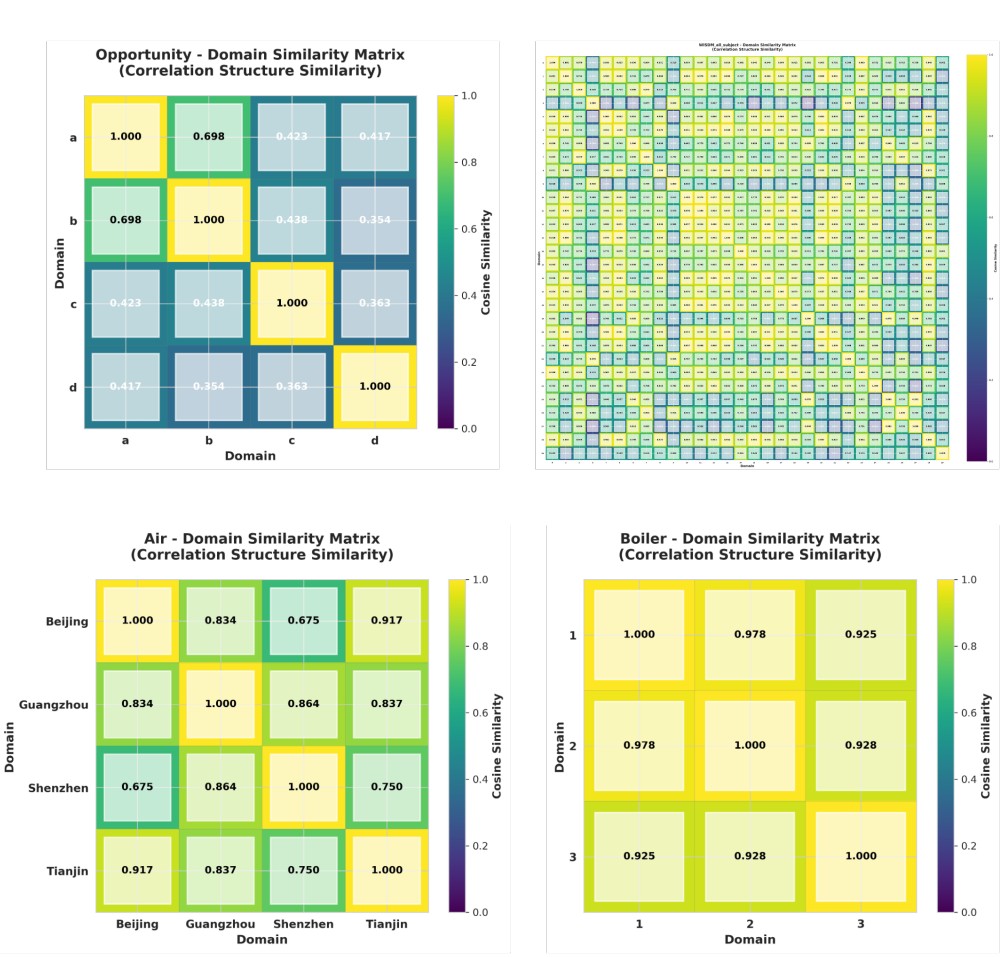

Figure 10: Heatmaps representing cross-domain similarity based on variable correlation on four datasets: Opportunity, WISDM, Air quality, and Boiler.

Table 9: Comparison of optimal-transport-based UDA methods.

| Method | OT granularity | Modality | Task |
|---|---|---|---|
| MLOT (Kerdoncuff et al., 2020a) | Instance-level | Generic vector/image | UDA |
| OTDA (Courty et al., 2016a) | Instance-level | Generic vector/image | UDA |
| DAEVS (Aritake & Hino, 2022a) | Instance-level | Generic vector/sensor (target-only extra features) | UDA with dimensional mismatch |
| RAINCOAT (He et al., 2023) | Instance-level | Timeseries | UDA w/wo label shift |
| IDSA (Ours) | Sensor-level | Multivariate timeseries | UDA with sensor-wise domain shift |

their correlation matrices. For each dataset, we report the mean, standard deviation, and min–max of $\text{sim}(i, j)$ over all unordered domain pairs in Table 8. Wearable datasets, Opportunity and WISDM datasets, show substantial drift: **Opportunity** mean similarity $\approx 45\%$ (113 channels), **WISDM** $\approx 67\%$ with high variance (3 channels), whereas non-wearable **Air** $\approx 81\%$ and **Boiler** $\approx 94\%$ are more stable. These results justify sensor-level transport rather than assuming one invariant structure. The overall results, including all cross-domain similarity based on each domain's own variable-wise correlation matrix per all four datasets, are visualized in Figure 10

### A.7.2 DISCUSSION WITH OPTIMAL TRANSPORT-BASED METHODS

There is a line of works adopting optimal transport (OT) for domain adaptation (Kerdoncuff et al., 2020a; Courty et al., 2016a; Aritake & Hino, 2022a). Table 9 summarizes the conceptual comparison across prior OT-based domain adaptation methods and IDSA. OTDA (Courty et al., 2016a) and MLOT (Kerdoncuff et al., 2020a) align sample distributions in a feature space whose coordinates are assumed to have stable semantics across domains. DAEVS (Aritake & Hino, 2022a) addresses a heterogeneous setting where the target has extra features, applying OT on the shared part while modeling the extra part with pseudo-labeling. RAINCOAT (He et al., 2023), which we already include as a baseline, aligns latent time and frequency features via Sinkhorn divergence and corrects possible label shift. In contrast, our IDSA treats each sensor channel as an entity and performs OT at the **sensor level** by learning a sensor channel-wise transport map to resolve sensor-wise domain shift, which is illustrated in our motivation. Thus, our approach differs from instance-level OT methods and instead addresses a distinct and commonly encountered problem in wearable MTS.

We additionally implement the MLOT (Kerdoncuff et al., 2020a) and OTDA (Courty et al., 2016a) models for more thorough performance comparison on OT-based DA methods, while excluding DAEVS (Aritake & Hino, 2022a), whose setting is different from ours, requiring additional target dimension. The results are presented in the Table 10. From the results, we can infer that Raincoat, which models OT specifically for time-series data, performs better than the other two traditional OT-based UDA methods. Our methodology, which models the transportation map for sensor-level optimal transport, still demonstrates the highest performance even when compared to newly adopted traditional OT methods.

### A.8 LIMITATION

While IDSA demonstrates strong performance in unsupervised domain adaptation for spatio-temporal sensor data, there are some limitations to consider. First, when each domain is associated with a very large number of sensors, the model introduces additional computational overhead due to the graph generation and the need to compute pairwise similarities for spatial transportation loss. This may limit scalability to very large datasets, and existing mini-batch training methods (Chen et al., 2018; Zheng et al., 2022) could be leveraged to mitigate the burden. Second, IDSA contains several hyperparameters that may require dataset-specific tuning for optimal performance. Developing more adaptive or automated hyperparameter selection techniques could improve usability. There are risks of compromising privacy if the model enables the re-identification of individuals from anonymized sensor data. Rigorous testing and the development of appropriate safeguards would be necessary before deployment in high-stakes scenarios. Despite these limitations and with responsible usage, we believe IDSA provides a valuable tool for learning robust sensor representations across domains to enable label-efficient yet accurate downstream predictions.

Table 10: Performance comparison with OT baselines on HAR datasets (Opportunity, WISDM).

| S → T | Metric (%) | Models | | | | |
|---|---|---|---|---|---|---|
| | | MLOT | OTDA | RAINCOAT | IDSA + Deep Coral | IDSA + CLUDA |
| | | **Opportunity** | | | | |
| 1 → 2 | Acc. | 74.30 | 70.39 | 82.89 | 86.82 | 86.03 |
| | F1 | 76.66 | 66.25 | 83.56 | 88.69 | 88.09 |
| 1 → 3 | Acc. | 69.25 | 69.88 | 77.23 | 81.06 | 86.96 |
| | F1 | 42.32 | 42.58 | 58.02 | 63.41 | 88.11 |
| 1 → 4 | Acc. | 72.39 | 75.64 | 75.39 | 82.57 | 84.45 |
| | F1 | 79.96 | 68.85 | 58.99 | 77.11 | 88.89 |
| 2 → 1 | Acc. | 83.38 | 75.93 | 81.64 | 89.30 | 85.96 |
| | F1 | 85.92 | 72.24 | 65.09 | 90.95 | 88.38 |
| 2 → 3 | Acc. | 70.19 | 69.88 | 77.73 | 81.37 | 84.78 |
| | F1 | 65.05 | 42.58 | 58.45 | 79.42 | 87.15 |
| 2 → 4 | Acc. | 73.46 | 70.95 | 71.48 | 74.27 | 81.23 |
| | F1 | 81.81 | 67.79 | 75.45 | 80.63 | 87.45 |
| 3 → 1 | Acc. | 67.62 | 67.91 | 85.54 | 83.38 | 79.08 |
| | F1 | 40.07 | 40.37 | 63.83 | 63.32 | 62.42 |
| 3 → 2 | Acc. | 65.08 | 62.57 | 78.52 | 81.01 | 80.45 |
| | F1 | 40.32 | 42.20 | 60.28 | 62.23 | 61.62 |
| 3 → 4 | Acc. | 56.84 | 55.50 | 58.20 | 80.43 | 82.84 |
| | F1 | 41.90 | 50.97 | 64.00 | 86.67 | 88.09 |
| 4 → 1 | Acc. | 70.20 | 72.49 | 87.89 | 83.38 | 93.69 |
| | F1 | 58.69 | 58.62 | 88.32 | 83.98 | 94.71 |
| 4 → 2 | Acc. | 68.99 | 65.92 | 75.00 | 76.82 | 82.12 |
| | F1 | 50.01 | 40.99 | 61.56 | 76.62 | 84.66 |
| 4 → 3 | Acc. | 69.88 | 69.88 | 69.14 | 76.71 | 88.51 |
| | F1 | 59.15 | 59.11 | 72.19 | 69.83 | 89.39 |
| **Avg.** | Acc. | 70.13 | 68.91 | 76.72 | 81.44 | 84.68 |
| | F1 | 60.16 | 54.38 | 67.48 | 76.91 | 84.08 |
| | | **WISDM** | | | | |
| 2 → 28 | Acc. | 82.22 | 80.00 | 83.15 | 86.67 | 86.67 |
| | F1 | 59.60 | 81.52 | 76.62 | 78.33 | 83.47 |
| 7 → 2 | Acc. | 65.85 | 65.85 | 78.05 | 68.30 | 82.93 |
| | F1 | 52.01 | 50.84 | 56.68 | 55.20 | 73.22 |
| 7 → 26 | Acc. | 73.17 | 73.73 | 73.17 | 78.05 | 73.17 |
| | F1 | 38.96 | 36.27 | 55.31 | 46.76 | 39.40 |
| 12 → 7 | Acc. | 64.58 | 64.58 | 79.17 | 83.33 | 81.25 |
| | F1 | 55.14 | 71.22 | 69.21 | 73.10 | 79.27 |
| 12 → 19 | Acc. | 54.55 | 71.21 | 46.97 | 54.54 | 80.30 |
| | F1 | 41.38 | 67.81 | 31.82 | 40.52 | 74.93 |
| 18 → 20 | Acc. | 41.46 | 36.59 | 76.83 | 68.29 | 82.93 |
| | F1 | 32.84 | 36.53 | 64.65 | 43.89 | 78.58 |
| 19 → 2 | Acc. | 58.54 | 51.22 | 63.41 | 63.41 | 65.85 |
| | F1 | 51.94 | 51.60 | 63.48 | 62.34 | 47.46 |
| 28 → 20 | Acc. | 70.73 | 75.61 | 85.37 | 97.56 | 87.80 |
| | F1 | 65.51 | 54.83 | 81.14 | 97.76 | 85.37 |
| 26 → 2 | Acc. | 63.31 | 53.66 | 67.07 | 80.50 | 90.24 |
| | F1 | 48.93 | 35.63 | 52.48 | 82.37 | 73.89 |
| 28 → 2 | Acc. | 60.98 | 63.41 | 87.80 | 75.61 | 90.24 |
| | F1 | 48.93 | 52.76 | 72.87 | 52.16 | 81.11 |
| **Avg.** | Acc. | 63.54 | 63.59 | 74.10 | 75.61 | 82.14 |
| | F1 | 49.52 | 53.90 | 62.43 | 63.24 | 71.67 |

