# OpenReview forum: "Inter-Domain Sensor Alignment for Unsupevised Domain Adaptation of Wearable Multivariate Time Series"
_ICLR.cc/2026/Conference — Submitted to ICLR 2026_

### Official Review · Reviewer_E2qa · 2025-10-28

**Soundness:** 2
**Presentation:** 2
**Contribution:** 2
**Rating:** 2
**Confidence:** 3

**Summary:**

This paper tackles the Unsupervised domain adaptation (UDA) problem for multivariate time-series in wearable sensor domain.
It motivates from the sensor-wise domain shift, where sensor placement/orientation or other factors might change for each individual in the same domain. Furthermore, it mentioned the potential intra-domain noisy or redundant information in each domain.

Based on two above motivations, it proposes Inter-Domain Sensor Alignment (IDSA) with two modules: (1) Inter-domain sensor alignment and (2) Intra-domain Sensor Decorrelation. (1) is implemented to align with the distance correlation of sensors, while (2) introduces a self-similarities w.r.t. identity matrix loss.

Experiments are conducted on activity recognition and surface electromyography datasets. Several methods are compared, with two being selected as the baseline to plug the proposed IDSA. In general, IDSA improves the two selected baseline methods by a big margin.

**Strengths:**

## Strengths
- The motivation is reasonable and well discussed and investigated, e.g., Sensor-wise Domain Shift can be demonstrated in Fig. 2. And with the motivation this paper defines the Sensor-wise Domain Shift as variations in sensor configurations, such as differences in sensor placement or orientation across domains.
- A inter-domain alignment module is designed to solve the Sensor-wise Domain Shift in Eq. (2).
- A Channel Decorrelation Loss is introduced to compress redundant or noisy information.
- Built on two baseline methods, the proposed model improves them by a big margin.

**Weaknesses:**

## Weaknesses
- Although the motivation is reasonable and well illustrated, the solution is relatively straightforward and has been explored intensively elsewhere.
> E.g., Optimal transport for domain adaptation has been widely-used, such as [R1-R3] and to name a few. No relevant papers are discussed or compared either in Related Work or Experiments.
> The idea of using distance among sensors as guideline or matching target is also well studied in different areas such as traffic flow prediction [R4].
- The introduction of Channel Decorrelation Loss is simply repelling the self-similarities, which is also a common technique in graph [R6] or contrastive learning [R5].
- Other modules are simply graph neural networks.
- Experimental baseline are relatively out-dated methods from 2023 or earlier. The proposed modules are applied to two baseline methods Deep Coral (Sun & Saenko, 2016) and CLUDA (Ozyurt et al., 2023), which are from 2016 and 2023. No optimal-tranport based UDA method is compared, which is highly relevant to this paper.
- No visualization of the transport map or self-correlation matrix is shown.


[R1] Kerdoncuff, Tanguy, Rémi Emonet, and Marc Sebban. "Metric learning in optimal transport for domain adaptation." International joint conference on artificial intelligence. IJCAI, 2020.

[R2] Courty, Nicolas, et al. "Optimal transport for domain adaptation." IEEE transactions on pattern analysis and machine intelligence 39.9 (2016): 1853-1865.

[R3] Aritake, Toshimitsu, and Hideitsu Hino. "Unsupervised domain adaptation for extra features in the target domain using optimal transport." Neural Computation 34.12 (2022): 2432-2466.

[R4] Zheng, Chuanpan, et al. "Gman: A graph multi-attention network for traffic prediction." Proceedings of the AAAI conference on artificial intelligence. Vol. 34. No. 01. 2020.

[R5] Zbontar, Jure, et al. "Barlow twins: Self-supervised learning via redundancy reduction." International conference on machine learning. PMLR, 2021.

[R6] Ma, Yuchen, Yanbei Chen, and Zeynep Akata. "Distilling knowledge from self-supervised teacher by embedding graph alignment." arXiv preprint arXiv:2211.13264 (2022).

**Questions:**

What is the difference of this method compared with OT-based UDA method?

What does the transport map or self-correlation matrix look?

Others please see above Weaknesses.

---

> ### Author Response · Authors · 2025-11-21
>
> **Contribution Clarification:**
> We thank the reviewer for acknowledging the motivation and also appreciate the feedback about the concerns regarding straightforward solutions.
> However, we emphasize the contribution of our work compared to existing methods in three perspectives, which are:
>
> * **Sensor-level OT, not instance-level OT.** We formulate an inter-domain transport over sensor channels and prove our spatial transportation loss is equivalent (up to a constant) to discrete $W\_{1}$ at the sensor level. The learned transport map $\textbf{A}\_{st}$ acts on data ($\hat{\textbf{X}} = \textbf{A}\_{st}\tilde{\textbf{X}}$) to resolve sensor-wise semantic mismatch that arises from placement, orientation, and subject effects before the feature extractor, which is orthogonal to prior OT-based domain adaptation methods.
> Most existing OT-based domain adaptation methods focus on data instance-level OT, while our method focuses on sensor-level OT. This point is discussed in detail below.
>
> * **Synergy from inter‑domain transport + intra‑domain decorrelation** The decorrelation term not only sparsifies intra‑domain graphs and reduces spurious cross‑channel couplings, but generates more informative target domain representation when optimized with the spatial transportation loss (information‑bottleneck view). This point is discussed in detail below.
>
> * **Plug‑in, practical gains.** As a plug‑in to standard UDA backbones, this sensor‑level OT yields consistent improvements on cross-subject and cross-repetition domain adaptation tasks, and remains robust in sensor‑permutation stress tests.

---

> ### Author Response · Authors · 2025-11-21
>
> **Discussion on OT-based domain adaptation methods:**
>
> We appreciate the reviewer’s suggestion of a more thorough discussion on OT-based DA, comparing our method against prior works([1], [2], [3]) provided by the reviewer, for highlighting our contribution in the context of optimal transport.
> However, before discussing the details of those works, we want to kindly remind the reviewer that, as one of the baselines of our work is an optimal transport-based domain adaptation method (Raincoat [4]), we did not completely exclude OT-based domain adaptation in the analysis.
> The table below represents a comparison across the mentioned OT-based methods, including ours.
>
>
> |       | MLOT [1] | OTDA [2]  |  DAEVS [3]  | RAINCOAT [4]  | Ours  |
> |---------|--------|-------|-------|-------|-------|
> |  OT granularity   | Instance-level  |  Instance-level  | Instance-level  | Instance-level | Sensor-level (channel-wise) |
> |  Modality    | Generic vector/image  | Generic vector/image | Generic vector/sensor (target-only extra features) | Timeseries | Multivariate timeseries |
> |  Task    | UDA  | UDA  | UDA with dimensional mismatch | UDA  w/wo label shift| UDA with sensor-wise domain shift |
> | OT objective              | $W\_{1}$ with learned Mahalanobis metric    | $W\_{1}$ / Sinkhorn (EMD/Sinkhorn)                    | $W\_{1}$ on shared features + extra-feature modeling               | Sinkhorn divergence (entropic OT)                   | Discrete $W\_{1}$ at sensor level (proved equivalence)       |
>
>
>
> While all of these methods leverage optimal transport, they operate at different levels and with different assumptions. OTDA and MLOT align sample distributions in a feature space whose coordinates are assumed to have stable semantics across domains. DAEVS addresses a heterogeneous setting where the target has extra features, applying OT on the shared part while modeling the extra part with pseudo-labeling. RAINCOAT, which we already include as a baseline, aligns latent time and frequency features via Sinkhorn divergence and corrects possible label shift. In contrast, our IDSA treats each sensor channel as an entity and performs OT at the **sensor level** by learning a sensor channel-wise transport map to resolve sensor-wise domain shift, which is illustrated in our motivation, and coupling it with intra-domain decorrelation. Thus, our approach is complementary to instance-level OT methods and targets a different and common motivating problem in wearable MTS.
>
> We additionally implement the MLOT and OTDA models for more thorough performance comparison on OT-based DA methods, while excluding DAEVS, whose setting is different from ours, requiring additional target dimension. The results are presented in the table below.
>
> |               | WISDM Acc.| WISDM F1. | Opportunity Acc. |Opportunity F1. |
> |---------------|---------|---------|---------|---------|
> |  MLOT [1]     |  63.54  |  49.52  |  70.13  |  60.16  |
> |  OTDA [2]     |  63.59  |  53.90  |  68.91  |  54.38  |
> |  Raincoat [4] |  74.10  |  62.43  |  76.72  |  67.47  |
> |  Ours         |  75.61  |  63.24  |  81.44  |  76.91  |
>
> From the results, we can infer that Raincoat, which models OT specifically for timeseries data, performs better than the other two traditional OT-based UDA methods. Our methodology, which models the transportation map for sensor-level optimal transport, still demonstrates the highest performance even when compared to newly adopted traditional OT methods. Full results of the experiment, including each performance scenario, are available in Table 9 in the Appendix of the revised manuscript.
>
> [1] Kerdoncuff, Tanguy, Rémi Emonet, and Marc Sebban. "Metric learning in optimal transport for domain adaptation." International Joint Conference on Artificial Intelligence (IJCAI), 2020.
>
> [2] Courty, Nicolas, et al. "Optimal transport for domain adaptation."  IEEE Transactions on Pattern Analysis and Machine Intelligence 39.9 (2016): 1853–1865.
>
> [3] Aritake, Toshimitsu, and Hideitsu Hino. "Unsupervised domain adaptation for extra features in the target domain using optimal transport." Neural Computation 34.12 (2022): 2432–2466.
>
> [4] Purushotham, Sanjay, Zhaowei Wang, and Yan Liu.
> "Domain Adaptation for Time Series Under Feature and Label Shifts."
> Proceedings of the 28th ACM SIGKDD Conference on Knowledge Discovery and Data Mining (KDD), 2022.

---

> ### Author Response · Authors · 2025-11-21
>
> **Novelty of Decorrelation Loss:**
> We acknowledge that the proposed decorrelation loss function and its role are similar to those discussed in the papers mentioned by the reviewer. However, we emphasize that when combined with the spatial transportation loss, the decorrelation loss encourages the learned representations to satisfy the information bottleneck principle: the target representation retains high mutual information with the source distribution while discarding target-specific domain information, thereby facilitating more effective domain adaptation.
>
> Under the assumption that the source data instance and target data instance have identical classes, the synergy between the decorrelation loss and our proposed loss can be interpreted through an information-bottleneck perspective.
>
> > [Theorem] Let the source-wise information bottleneck be defined as: ${IB}\_{s} = I(\textbf{Z}\_{t}, {X}\_{s}) - \beta I(Tar, \textbf{Z}\_{t}),$ where ${X}\_s$ indicates the source distribution and $Tar$ denotes target domain-specific information. In particular, if IDSA is optimized through loss functions, the transformed target representation satisfies the source-wise information bottleneck. $\min ({L}\_{\text{st}} + {L}\_{\text{dec}}) \Rightarrow \max {IB}\_{s}.$
>
> Our transport plan enforces each target instance to align its sensor-wise distribution with that of the corresponding source instance, effectively encouraging $ \textbf{Z}\_{s} \approx \textbf{Z}\_t $.
> When combined with the decorrelation loss, this objective promotes target representations $\textbf{Z}\_t$ that preserve high mutual information with the source distribution while suppressing mutual information with target-specific domain factors $Tar$. Consequently, the learned representation becomes well-suited for domain adaptation. We appreciate the reviewer’s insightful comment, and additional theoretical discussion is provided in the revised paper.

---

> ### Author Response · Authors · 2025-11-21
>
> **Experiments with additional latest baselines:**
> In addition to newly adopted OT-based DA baselines following the reviewer's suggestion, we ran UniMTS, a recent pretrain to finetune approach for motion time-series.
> To evaluate the effectiveness of our method in a domain adaptation setting, we compared its performance with UniMTS on two HAR datasets. For fairness, we first fine-tuned UniMTS using its original procedure and then further fine-tuned it on the source-domain training data. Although UniMTS achieves competitive results in a few scenarios, our approach yields superior performance in most cases, suggesting that direct distributional transformation is more effective than augmentation-based handling of sensor-configuration variability, as shown in the table below. For the Opportunity dataset, UniMTS's relatively low performance is expected, as it uses only IMU sensors, thereby limiting the amount of usable input data compared to ours, which uses the full set of available sensors.
>
> ### WISDM Results
>
> | Setting | UniMTS-Acc | UniMTS-F1 | Ours-Acc | Ours-F1 |
> |--------|-------------|------------|-----------|----------|
> | 2 → 28 | 77.77 | 73.36 | 86.67 | 78.33 |
> | 7 → 2 | 90.24 | 81.11 | 82.93 | 73.22 |
> | 7 → 26 | 58.54 | 65.05 | 78.05 | 46.75 |
> | 12 → 7 | 89.58 | 89.88 | 83.33 | 73.10 |
> | 12 → 19 | 89.39 | 89.83 | 80.30 | 74.93 |
> | 18 → 20 | 53.66 | 57.36 | 82.93 | 78.58 |
> | 19 → 2 | 85.37 | 74.86 | 65.85 | 47.46 |
> | 28 → 20 | 73.17 | 61.01 | 97.56 | 97.56 |
> | 26 → 2 | 95.12 | 91.53 | 90.24 | 73.89 |
> | 28 → 2 | 70.73 | 65.56 | 90.24 | 81.11 |
>
> ### Opportunity Results
>
> | Setting | UniMTS-Acc | UniMTS-F1 | Ours-Acc | Ours-F1 |
> |---------|-------------|------------|-----------|----------|
> | 1 → 2 | 61.73 | 64.89 | 86.82 | 88.69 |
> | 1 → 3 | 74.54 | 75.86 | 86.96 | 88.11 |
> | 1 → 4 | 69.17 | 74.94 | 84.45 | 88.89 |
> | 2 → 1 | 81.37 | 83.12 | 89.30 | 90.95 |
> | 2 → 3 | 67.08 | 42.96 | 84.78 | 87.15 |
> | 2 → 4 | 58.71 | 59.43 | 81.23 | 87.45 |
> | 3 → 1 | 70.49 | 55.41 | 83.38 | 63.32 |
> | 3 → 2 | 61.73 | 49.48 | 81.01 | 62.23 |
> | 3 → 4 | 58.98 | 48.10 | 82.84 | 88.09 |
> | 4 → 1 | 47.85 | 40.92 | 93.69 | 94.71 |
> | 4 → 2 | 46.09 | 42.66 | 82.12 | 84.66 |
> | 4 → 3 | 64.91 | 49.05 | 88.51 | 89.39 |

---

> ### Author Response · Authors · 2025-11-21
>
> **Visualization of inter-domain transportation map and intra-domain sensor correlation**
> We thank the reviewer for requesting visualizations of the learned transportation map and intra-domain sensor relations. Following this suggestion, we additionally visualize (i) transportation map as inter-domain bipartite graph between source and target domain sensors, where edge thickness encodes the learned weights of $A\_{ST}$ and (ii) intra-domain self correlation heatmap. To illustrate robustness of transportation plan under sensor misalignment, we extend the transportation map visualization on sensor-wise swap (permutation) setting. The result reveals that, after the swap, the learned transport reassigns mass to the correct counterpart while connections for non‑swapped sensors remain stable. The self‑correlation matrices show that the decorrelation loss suppresses spurious off‑diagonal couplings while preserving per‑channel identity. We added the visualizations to the revised version of our paper (Appendix).

---

> ### Comment · Reviewer_E2qa · 2025-11-26
>
> Thank authors for the detailed rebuttal.
>
> The additional OT baselines enriched the comparison. While the comparison with UniMTS is less convincing as below:
> > For the WISDM results, it looks quite mixed, sometimes UniMTS wins sometimes the proposed wins.
> > While for Opportunity:
> "For the Opportunity dataset, UniMTS's relatively low performance is expected, as it uses only IMU sensors, thereby limiting the amount of usable input data compared to ours, which uses the full set of available sensors." >> This seems like an unfair comparison.
>
> Therefore, I am not convinced by the rebutall and will remain my score.

---

> > ### Author Response · Authors · 2025-11-27
> >
> > ## Follow-up Clarification on UniMTS Comparison and Fairness of the Setting
> > We thank the reviewer for pointing out the UniMTS comparison result and the potential unfairness in our original description. We agree that our previous explanation for the Opportunity results ("UniMTS uses only IMU sensors, whereas ours uses all sensors") could be interpreted as an unfair comparison. So we clarified the experimental setting with a more detailed explanation and also extended previous comparison by adding our performance result in the new setting (only IMU sensor signals) to address this.
> >
> > **Regarding WISDM setting and performance comparison:**
> >
> > The sensor signals are fairly given to both UniMTS, the pretrain-finetune framework that is designed and evaluated primarily on IMU-based motion time-series to handle variability of IMU sensors, and ours, because WISDM only contains variables as IMU sensor channels. In addition, its pretraining stage leverages **textual activity descriptions** as an **extra modality**, which our method does **not** use. In our WISDM experiments, we thus follow the official UniMTS protocol (sensor signals + text during pretraining), while our method uses only sensor signals. We consider that this experimental setting, where UniMTS has access to additional information, may explain the "mixed performance" observed between UniMTS and our method on the WISDM results.
> >
> > **Regarding Opportunity setting and fairness:**
> >
> > We appreciate the reviewer’s observation that comparing our method trained with the full sensor set against UniMTS with only IMU inputs is unfair, even though UniMTS is inherently designed for IMU signals and further leverages additional textual descriptions. To provide a more controlled comparison, we have included an additional column reporting the performance of our model when restricted to IMU signals only, thereby matching the UniMTS input configuration.
> > The updated results shown below demonstrate that our IMU-only variant consistently matches or exceeds the performance of UniMTS across the majority of source→target transfer settings on the Opportunity dataset, yielding an average improvement of 18.91\%. Notably, this performance advantage is achieved despite UniMTS utilizing an additional modality (text encoder) and benefitting from pretraining with synthetic dataset. These findings further highlight the distinctiveness and effectiveness of our approach, underscoring that its contributions are not merely additive but represent a fundamentally stronger modeling strategy for domain-generalized human activity recognition.
> >
> > **Opportunity Results**
> >
> > | Setting | UniMTS-Acc | UniMTS-F1 | Ours(IMU)-Acc | Ours(IMU)-F1 |Ours-Acc | Ours-F1 |
> > |---------|-------------|------------|-----------|----------|---|---|
> > | 1 → 2 | 61.73 | 64.89 | 77.65 | 81.77 |86.82 | 88.69 |
> > | 1 → 3 | 74.54 | 75.86 | 79.50 | 66.61 |86.96 | 88.11 |
> > | 1 → 4 | 69.17 | 74.94 | 73.19 | 77.47 |84.45 | 88.89 |
> > | 2 → 1 | 81.37 | 83.12 | 82.80 | 84.76 |89.30 | 90.95 |
> > | 2 → 3 | 67.08 | 42.96 | 67.39 | 39.67 |84.78 | 87.15 |
> > | 2 → 4 | 58.71 | 59.43 | 65.15 | 48.75 |81.23 | 87.45 |
> > | 3 → 1 | 70.49 | 55.41 | 86.82 | 90.54 |83.38 | 63.32 |
> > | 3 → 2 | 61.73 | 49.48 | 73.46 | 72.69 |81.01 | 62.23 |
> > | 3 → 4 | 58.98 | 48.10 | 65.15 | 52.80 |82.84 | 88.09 |
> > | 4 → 1 | 47.85 | 40.92 | 81.95 | 83.25 |93.69 | 94.71 |
> > | 4 → 2 | 46.09 | 42.66 | 76.82 | 55.32 |82.12 | 84.66 |
> > | 4 → 3 | 64.91 | 49.05 | 77.02 | 76.21 |88.51 | 89.39 |
> >
> > For the WISDM results, where UniMTS and our method show "mixed performance", they were based on a setting where UniMTS had access to strictly more information than ours (IMU + textual label information vs. IMU-only).
> >
> > We are grateful for your feedback, as it helped us improve both the clarity and thoroughness of our experimental analysis, and we have updated the revision accordingly to make this point explicit.
> > We hope this clarifies the reviewer’s additional concerns, and we are happy to provide further discussion if needed.

---

### Official Review · Reviewer_vQxH · 2025-10-29

**Soundness:** 3
**Presentation:** 3
**Contribution:** 2
**Rating:** 6
**Confidence:** 4

**Summary:**

In this study, the authors design a plug-in module for unsupervised domain adaptation on sensor data. The application scenarios tested include human activity recongiont and sEMG. On the technical part, the key component of this paper is the plug-in module that does not need to change existing model architecture. The experiments are conducted on 5 real-world datasets, and the authors have demonstrated the performance gain compared to multiple baseline models. Overall, the problem studied is interesting, but there are concerns that need to be addressed carefully. Please see the following comments in detail.

**Strengths:**

[1] The topic studied in this paper is interesting, and could have broad applications in the real world.

[2] Most figures are well-designed, which benefits the understanding of this paper. The reviewer appreciates the efforts from the authors.

[3] The experiments are conducted on two tasks with multiple real-world datasets.

**Weaknesses:**

In the abstract, the authors mentioned the improvement of the WISDM dataset. The reviewer understands this emphasis is trying to highlight the effectiveness of the proposed method. However, it might mislead the readers in the future. Since this is not the only dataset used in this study. Also, the improvement on this dataset might not represent the improvements on all datasets tested by the proposed method.

The last sentence in the opening paragraph of the intro is hard to understand.

In line 50, does the phrase “various domains” has the same meaning like source and target domains?

What is the “MTS” data in line 53. It would be great to explain all abbreviations when they first appear.

Figure 2 introduces many concerns. First, it is challenging to understand the whole setting. Also, if the sensor here refers to IMU sensor, there are multiple channels for each IMU sensor, how does the “similarity” actually calculated remains unclear. Third, the sensors could rotate and different channels or sensors from source and target at the same position might have significant differences in terms of reading, this is intuitive. In line 168, the authors mentioned the “functuional consistency”, which is not easy to understand.

In the appendix, could you explain why there are only 4 subjects and 4 classes of activities considered for this study? It seems the task has little challenge.

For baselines, while there are many advanced activity recognition models in recent years, the baselines selected are general models for time series, which might raise concerns about the technical advancement.  Here are some recent examples of human activity recognition: "NeurIPS 2024, UniMTS: Unified Pre-training for Motion Time Series", "Ubicomp 2024, CrossHAR: Generalizing Cross-dataset Human Activity Recognition via Hierarchical Self-Supervised Pretraining".

It will be easier for readers if the tasks of experiment could be introduced  in the main content. For example, what are the input and output.

The writings could be further enhanced. For example, in the opening para of Section 3. This long sentence makes it difficult to understand.

In line 816, the color could be changed back to black.

**Questions:**

In the abstract, the authors mentioned the improvement of the WISDM dataset. The reviewer understands this emphasis is trying to highlight the effectiveness of the proposed method. However, it might mislead the readers in the future. Since this is not the only dataset used in this study. Also, the improvement on this dataset might not represent the improvements on all datasets tested by the proposed method.

The last sentence in the opening paragraph of the intro is hard to understand.

In line 50, does the phrase “various domains” has the same meaning like source and target domains?

What is the “MTS” data in line 53. It would be great to explain all abbreviations when they first appear.

Figure 2 introduces many concerns. First, it is challenging to understand the whole setting. Also, if the sensor here refers to IMU sensor, there are multiple channels for each IMU sensor, how does the “similarity” actually calculated remains unclear. Third, the sensors could rotate and different channels or sensors from source and target at the same position might have significant differences in terms of reading, this is intuitive. In line 168, the authors mentioned the “functuional consistency”, which is not easy to understand.

In the appendix, could you explain why there are only 4 subjects and 4 classes of activities considered for this study? It seems the task has little challenge.

For baselines, while there are many advanced activity recognition models in recent years, the baselines selected are general models for time series, which might raise concerns about the technical advancement.  Here are some recent examples of human activity recognition: "NeurIPS 2024, UniMTS: Unified Pre-training for Motion Time Series", "Ubicomp 2024, CrossHAR: Generalizing Cross-dataset Human Activity Recognition via Hierarchical Self-Supervised Pretraining".

It will be easier for readers if the tasks of experiment could be introduced  in the main content. For example, what are the input and output.

The writings could be further enhanced. For example, in the opening para of Section 3. This long sentence makes it difficult to understand.

In line 816, the color could be changed back to black.

---

> ### Author Response · Authors · 2025-11-21
>
> **Weakness 1 (Modification of Abstract):** Thanks for the suggestion. We modified the abstract to reduce the confusion the original version may have induced.
>
> **Weakness 2 (Modification of Introduction):** We appreciate the reviewer’s feedback. In response, we have revised the final sentence of the first introductory paragraph to: In practice, changes in placement, orientation of sensors induce \emph{sensor-wise domain shift} across different domains, which is distinct from the usual temporal covariate shift and remains relatively underexplored.
> We hope this revision improves the clarity of the original statement.
>
> **Weakness 3 (Figure 2 Clarification):**
> We apologize for any confusion caused by the unclear terminology. Throughout the paper, the term sensor refers to a sensor unit consisting of a single channel. Under this definition, an IMU sensor comprises six signals (three from the accelerometer and three from the gyroscope). We will revise the manuscript to explicitly clarify this definition. Accordingly, the similarity of each sensor channel is computed using cosine similarity, as illustrated in Figure 2. Moreover, when sensors are perfectly aligned across domains, we expect their functional behavior to remain consistent. However, achieving perfect alignment is practically infeasible, giving rise to sensor-wise domain shifts, which is an issue that motivates the development of our method.
>
> **Weakness 4 (Experimental Setting of Opportunity HAR dataset):** The Opportunity HAR dataset comprises signals from 113 sensors placed at various locations on the body. It contains recordings from **4 subjects**, each of which is treated as a separate domain. The dataset provides two levels of label annotation: (1) locomotion, representing low-level activities such as sitting, standing, walking, and lying; and (2) gestures, representing 17 higher-level actions. Following prior work[1], we focus on the locomotion label annotations, resulting in a **4-class** classification setting.
>
> [1] Sensor alignment for multivariate time-series unsupervised domain adaptation, AAAI 2023

---

> ### Author Response · Authors · 2025-11-21
>
> **Weakness 5 (Baselines):** We thank the reviewer for suggesting relevant baselines[2],[3]. Both methods follow a pre-training–fine-tuning paradigm, where the pre-training stage resembles domain generalization and the fine-tuning stage corresponds to domain adaptation. These approaches rely on augmentation strategies to address sensor-configuration variability during pre-training and to enhance generalization to unseen domains. While such techniques are effective when the target domain is entirely unknown, we argue that our approach, which explicitly aligns the target distribution with the source distribution through a transformation matrix, is more appropriate for the domain adaptation setting, where partial information about the target domain is available.
>
> To evaluate the effectiveness of our method, we compared its performance with UniMTS[2] on two HAR datasets in a domain adaptation setting. For fairness, we first fine-tuned UniMTS using its original procedure and then further fine-tuned it on the source-domain training data. As shown in below WISDM results table, although UniMTS achieves competitive results in a few scenarios, our approach yields superior performance in most cases, suggesting that direct distributional transformation is more effective than augmentation-based handling of sensor-configuration variability. For the Opportunity dataset, the relatively low performance of UniMTS is expected, as the model utilizes only IMU sensors, thereby restricting the amount of usable input data compared to ours, which utilizes the whole set of available sensors. We also extend the main table by adding the UniMTS method in the revised manuscript.
>
> **WISDM Results**
>
> | Setting | UniMTS-Acc | UniMTS-F1 | Ours-Acc | Ours-F1 |
> |--------|-------------|------------|-----------|----------|
> | 2 → 28 | 77.77 | 73.36 | 86.67 | 78.33 |
> | 7 → 2  | 90.24 | 81.11 | 82.93 | 73.22 |
> | 7 → 26 | 58.54 | 65.05 | 78.05 | 46.75 |
> | 12 → 7 | 89.58 | 89.88 | 83.33 | 73.10 |
> | 12 → 19 | 89.39 | 89.83 | 80.30 | 74.93 |
> | 18 → 20 | 53.66 | 57.36 | 82.93 | 78.58 |
> | 19 → 2 | 85.37 | 74.86 | 65.85 | 47.46 |
> | 28 → 20 | 73.17 | 61.01 | 97.56 | 97.56 |
> | 26 → 2 | 95.12 | 91.53 | 90.24 | 73.89 |
> | 28 → 2 | 70.73 | 65.56 | 90.24 | 81.11 |
>
> **Opportunity Results**
>
> | Setting | UniMTS-Acc | UniMTS-F1 | Ours-Acc | Ours-F1 |
> |---------|-------------|------------|-----------|----------|
> | 1 → 2 | 61.73 | 64.89 | 86.82 | 88.69 |
> | 1 → 3 | 74.54 | 75.86 | 86.96 | 88.11 |
> | 1 → 4 | 69.17 | 74.94 | 84.45 | 88.89 |
> | 2 → 1 | 81.37 | 83.12 | 89.30 | 90.95 |
> | 2 → 3 | 67.08 | 42.96 | 84.78 | 87.15 |
> | 2 → 4 | 58.71 | 59.43 | 81.23 | 87.45 |
> | 3 → 1 | 70.49 | 55.41 | 83.38 | 63.32 |
> | 3 → 2 | 61.73 | 49.48 | 81.01 | 62.23 |
> | 3 → 4 | 58.98 | 48.10 | 82.84 | 88.09 |
> | 4 → 1 | 47.85 | 40.92 | 93.69 | 94.71 |
> | 4 → 2 | 46.09 | 42.66 | 82.12 | 84.66 |
> | 4 → 3 | 64.91 | 49.05 | 88.51 | 89.39 |
>
>
>
> [2] UniMTS: Unified Pre-training for Motion Time Series, Neurips 2024
>
> [3] CrossHAR: Generalizing Cross-dataset Human Activity Recognition via Hierarchical Self-Supervised Pretraining, Ubicomp 2024

---

> > ### Comment · Reviewer_vQxH · 2025-11-27
> >
> > Thank you for the detailed responses. For the baseline models that only utilize IMU data, is it possible that their performance will be increased when combined with other modalities?

---

> > > ### Author Response · Authors · 2025-11-27
> > >
> > > ## Follow-up Analysis on UniMTS Comparison and Additional Experiment
> > > We thank the reviewer for the thoughtful question regarding wether UniMTS may achieve further performance gains when additional sensor modalities are available. We would like to provide further clarification of the experimental setting for UniMTS with a more detailed explanation, as well as an extended comparison by adding our performance result under a new setting (only IMU sensor signals) for more thorough comparison between UniMTS and our method.
> > >
> > > **WISDM setting:**
> > >
> > > Because the WISDM dataset contains only IMU sensor channels, both UniMTS and our method operate under the same sensor modality in this setting. In addition, UniMTS adopts a pretraining stage leveraging textual activity descriptions as an extra modality, which our method does not use. When experimenting on the WISDM dataset, we follow the official UniMTS protocol (sensor signals + text during pretraining), while our method uses only sensor signals. Consequently, there are no additional sensor modalities that could be provided to UniMTS (or to our method) in the WISDM setting, and we consider that the result reflects the additional information in that UniMTS has access due to its text-based pretraining.
> > >
> > > **Opportunity setting and additional experiment:**
> > >
> > > For the Opportunity dataset, the reviewer raises a valid question about whether providing additional modalities available in the dataset to UniMTS could increase its performance.
> > > We agree that additional sensor modalities may lead to better performance, just as textual descriptions already do for UniMTS. However, as equations for embedding of sensor signal extraction in UniMTS are fundamentally based on a 3-axis IMU sensor signal input structure, the additional non-IMU sensor inputs needs extra consideration. Incorporating non-IMU sensor modalities into UniMTS would require additional encoders and architectural changes, and thus falls outside the scope of UniMTS's intended design. By contrast, our method naturally support heterogeneous sensor sets by not assuming a fixed IMU structure in sensor-level transportation.
> > >
> > > Furthermore, to provide a more thorough comparison between UniMTS and ours on the Opportunity dataset in a controlled setting, we conducted an additional experiment and included an additional column reporting the performance of our model when restricted to IMU signals only, thereby matching the UniMTS input configuration.
> > > The updated results demonstrate that our IMU-only variant consistently matches or exceeds the performance of UniMTS across the majority of source→target transfer settings on the Opportunity dataset, yielding an average improvement of 18.91\%. Notably, this performance advantage is achieved despite UniMTS utilizing an additional modality (text description in pretraining stage) and benefiting from pretraining. These findings further highlight the distinctiveness and effectiveness of our approach, underscoring that its contributions are not merely additive but represent a fundamentally stronger modeling strategy for domain-generalized human activity recognition.
> > >
> > > **Opportunity Results**
> > >
> > > | Setting | UniMTS-Acc | UniMTS-F1 | Ours(IMU)-Acc | Ours(IMU)-F1 |
> > > |---------|-------------|------------|-----------|----------|
> > > | 1 → 2 | 61.73 | 64.89 | 77.65 | 81.77 |
> > > | 1 → 3 | 74.54 | 75.86 | 79.50 | 66.61 |
> > > | 1 → 4 | 69.17 | 74.94 | 73.19 | 77.47 |
> > > | 2 → 1 | 81.37 | 83.12 | 82.80 | 84.76 |
> > > | 2 → 3 | 67.08 | 42.96 | 67.39 | 39.67 |
> > > | 2 → 4 | 58.71 | 59.43 | 65.15 | 48.75 |
> > > | 3 → 1 | 70.49 | 55.41 | 86.82 | 90.54 |
> > > | 3 → 2 | 61.73 | 49.48 | 73.46 | 72.69 |
> > > | 3 → 4 | 58.98 | 48.10 | 65.15 | 52.80 |
> > > | 4 → 1 | 47.85 | 40.92 | 81.95 | 83.25 |
> > > | 4 → 2 | 46.09 | 42.66 | 76.82 | 55.32 |
> > > | 4 → 3 | 64.91 | 49.05 | 77.02 | 76.21 |
> > >
> > >
> > >
> > > We sincerely appreciate the reviewer’s question, which helped us strengthen the fairness and completeness of our comparison with UniMTS.
> > > We hope these clarifications fully address the reviewer’s concern, and we would be glad to provide any further discussion if needed.

---

### Official Review · Reviewer_Ly5z · 2025-10-31

**Soundness:** 2
**Presentation:** 3
**Contribution:** 2
**Rating:** 2
**Confidence:** 5

**Summary:**

This paper proposes IDSA, an unsupervised domain adaptation module for wearable multi-sensor data. It targets sensor-wise domain shifts caused by differences in sensor placement and orientation. IDSA introduces a spatial transport loss, formulated as an optimal transport problem to align sensor spatial correlations across domains, and a channel decorrelation loss to reduce redundant intra-domain correlations. The module can be easily integrated into UDA backbones and achieves consistent performance gains on five HAR and sEMG benchmarks.

**Strengths:**

1. The paper addresses an important challenge in wearable sensing. Misalignment at the sensor level is meaningful and relevant to real-world deployment, which has attracted recent research interests.

2. The proposed module is a plug-in module that can be integrated into existing UDA pipelines, making it very practical.

3. The presentation is clear, figures are intuitive, and mathematical derivations are easy to follow.

**Weaknesses:**

1. While the paper presents the idea of “inter-domain sensor alignment” as new for multi-sensor time-series data, there is extensive prior literature on multivariate time-series domain adaptation using structurally similar techniques, such as correlation or covariance alignment (e.g., Time Series Domain Adaptation via Sparse Associative Structure Alignment, AAAI 2021, and related works). These works address domain shifts in the spatial dependencies of multivariate time series. It seems that the main difference between these works and the proposed method lies in the conceptual distinction between “multi-sensor” and “multi-variate” alignment, making the originality of this paper appear incremental. In addition, the paper in AAAI 2021 also considers sparsity.

2. The authors emphasize handling differences in sensor "placement or orientation", yet the experimental setup appears to use datasets with identical sensor configurations across subjects, which does not actually reflect such spatial misalignment. This weakens the empirical justification of the claimed contribution.

3. The paper is vague about the meaning of the channel dimension (N) in the learned representation Z. According to Section 5.1, the sensor embedding P has the same dimension N×T as the input MTS data, implying that each of the N channels corresponds to an axis or variable rather than a complete sensor unit, as each sensor usually has multiple axes. If this is correct, then the modeling operates at the channel/variable level rather than per sensor, which questions whether the alignment is truly “sensor-wise.”

4. It is unclear whether IDSA can handle non-identical sensor deployments, such as different positional configurations, or only applies to the same sensor deployment for source and targets.

**Questions:**

1. Is the proposed method limited to domains with identical sensor deployments (same sensor types, positions, and count)? If not, can the authors provide additional experiments or analysis demonstrating robustness to heterogeneous or partially misaligned sensor configurations?

2. In Section 5.1, the representation dimension N is stated to “match the input dimensions of MTS data.” Does N correspond to individual sensor channels or sensors as composite entities? This clarification is important because if each channel corresponds merely to one axis of a sensor, the problem reduces to conventional multivariate variable alignment.

3. Could the authors explicitly differentiate their formulation from existing multivariate time-series adaptation approaches that align correlation or covariance matrices of multiple variables? Beyond naming the entities “sensors,” what is the core modeling difference or advantage of IDSA in capturing inter-sensor relations compared to these works? And can this be evidenced by a thorough experimental comparison?

---

> ### Author Response · Authors · 2025-11-21
>
> **(Weakness 1 \& Question 3) Comparison Against Existing Methods that align Correlation or Covariance Matrices (1/2):**
> We appreciate the reviewer for bringing the references [1], which also exploit the relationships across variables (sensors) in domain adaptation. Comparing our work with these studies (including [2], as raised by another reviewer) helps position our approach by showing that the distinction does not arise from a conceptual difference between multi-variable and multi-sensor settings. While these studies and our method all consider relationships between variables (sensors), our problem setting, assumptions, and objectives differ in fundamental ways.
>
>
> | Method          | Domain examples  |   Variable set example |  Task  |
> |-----------------|-----------|--------------|--------|
> | SASA [1]        | Regions, Age-based groups  | (Blood Glucose, Glucagon, Insulin) |   Timeseries UDA |
> | GCA [2]         | Simulations, Regions | (Blood Glucose, Glucagon, Insulin) |  Semi-supervised timeseries forecasting DA |
> | IDSA (Ours)     | Subjects, Repetitions | IMU/sEMG sensor channels (channel 1, channel 2, ..., channel n)  |  Timeseries UDA|
>
> Given a variable set such as (Blood Glucose, Glucagon, Insulin) used in prior works, they are well-defined quantities whose correlation or covariance may stably hold across domains.
> Let $X\_1, X\_2, \ldots, X\_n$ denote the variables, and $\varphi$ denotes a feature extractor. In standard DA, one often has
>
> $P\_S\left( y \mid \varphi(X\_1, X\_2, \ldots, X\_n) \right) = P\_T\left( y \mid \varphi(X\_1, X\_2, \ldots, X\_t) \right)$.
>
> However, SASA[1] and GCA[2] further assume that the inter-variable structure is stable:
>
> $P\_S\left( y \mid \varphi(X\_1, X\_2, \ldots, X\_n) \right) \neq P\_T\left( y \mid \varphi(X\_1, X\_2, \ldots, X\_t) \right)$,
>
> $P\_S\left( C \mid \varphi(X\_1, X\_2, \ldots, X\_n) \right) \approx  P\_T\left( C \mid \varphi(X\_1, X\_2, \ldots, X\_n) \right)$,
>  where $C$ denotes their structural dependency.
>
> Under this assumption, their methods learn to extract the invariant structural dependency among variables across different domains, such as:
>
> * Extracting sparse associative structure with intra- and inter-variable attention mechanisms and aligning those structures [1], and
> * Leveraging domain invariant causal structures while also modeling domain-specific components [2].
>
> We agree that correlation‑ or causality‑based alignment methods such as SASA and GCA are well‑motivated and effective under the assumption that variables are well‑defined physical/physiological quantities and their inter‑variable relationships are approximately stable across domains. However, in wearable MTS, each "variable" is a sensor channel whose semantics can easily change, even if there are no intentional data changes, across subjects or repetitions due to changes in placement from reattachment or characteristics of subjects. This kind of scenario is illustrated in Figure 1 (Right).
> In other words, the wearable MTS domain introduces problem scenarios in which the assumptions underlying methods such as SASA and GCA may not hold.
> We assume that it can result in substantially different variable relations for source and target domains:
>
> $P\_S\left( C \mid \varphi(X\_1, X\_2, \ldots, X\_n) \right) \neq P\_T\left( C \mid \varphi(X\_1, X\_2, \ldots, X\_n) \right)$.
>
> We therefore introduce a channel-wise (variable-wise) transport map $\textbf{A}\_{st}$ and model:
>
> $P\_S\left( C \mid \varphi(X\_1, X\_2, \ldots, X\_n) \right) \approx  P\_T\left( C \mid \varphi(X\_1, X\_2, \ldots, X\_n), \textbf{A}\_{st} \right)$.
>
> With this revised assumption considering 'sensor-wise domain shift', our approach explicitly learns a cross-domain sensor correspondence, in contrast to correlation/covariance alignment that assumes a single shared structure. We model **inter-domain** variable relations, in addition to **intra-domain** variable relations, under the assumption that inter-domain variable relationships may be dynamic.
>
> [1] Time Series Domain Adaptation via Sparse Associative Structure Alignment, AAAI 2021.
>
> [2] Transferable Time-Series Forecasting Under Causal Conditional Shift, TPAMI 2024.

---

> ### Author Response · Authors · 2025-11-21
>
> **(Weakness 1 \& Question 3) Comparison Against Existing Methods that align Correlation or Covariance Matrices (2/2):**
> To assess whether the substantial cross-domain differences in variable correlation and covariance require an additional inter-domain transformation beyond simple correlation alignment, we conduct the following supplementary experiments on four datasets:Boiler, Air Quality (from prior works), WISDM, and Opportunity (our datasets).
>
> | Dataset | Domains | Variables | Mean Sim. | Std Sim. | Min–Max Sim. | Severity of Cross‑Domain Relation Drift |
> |---|---|---:|---:|---:|---:|---|
> | Opportunity | 4 subjects  | 113 | 44.89 | 11.57 | 35.44–69.83 | Severe  |
> | WISDM  | 30 subjects  | 3 | 66.61 | 18.71 | 40.19–88.06 | High variance |
> | Air quality | 4 cities  | 11 | 81.29 | 7.91 | 67.49–91.75 | Moderate (local conditions) |
> | Boiler | 3 conditions  | 20 | 94.38 | ~2.7 | 92.47–97.84| Benign (stable physical sensors) |
>
> For each domain, we aggregate channel-wise time-series into feature statistics (mean/std) and apply PCA to stabilize dimensionality and compute the inter-variable correlation matrix from this. Domain similarity inside a dataset is then defined as the cosine similarity between vectorized correlation matrices. Corresponding domain similarity heatmaps of all four datasets are available in the Appendix of the revised manuscript.
> These results show that, in the cross‑subject / cross‑repetition setting, wearable datasets (Opportunity, WISDM) exhibit substantial drift in inter‑variable (sensor) relationships, whereas non‑wearable datasets (Air, Boiler) are more stable. In particular, Opportunity shows low mean similarity across domains (≈45\%) and WISDM shows the second lowest with highly variable similarity (≈67\% with a large spread), indicating that a variable-wise correlation/covariance structure is often distinct across domains.  By contrast, Air quality (≈81\%) and Boiler (≈94\%) display higher cross‑domain similarity. These observations motivate the need for an additional inter‑domain transportation process at the variable (sensor) level rather than relying solely on correlation alignment that assumes a single invariant structure. The comprehensive results of this experiment, including experimental setup, visualization, and more thorough analyses, are now provided in **Appendix A.7.** of the revised paper.
>
> We also evaluated SASA on WISDM and Opportunity datasets under the scenarios used in our work for a direct comparison with IDSA. Our method consistently outperformed SASA in accuracy across both datasets. We will incorporate these comparison results, including F1 scores, into our revised manuscript.
>
> ### WISDM — SASA vs. Ours (Accuracy, \%)
>
> | Setting | SASA (Acc) | Ours (Acc) |
> |:--:|--:|--:|
> | 2→28 | 84.94 | 86.67 |
> | 7→2  | 71.25 | 82.93 |
> | 7→26 | 81.08 | 78.05 |
> | 12→7 | 67.30 | 83.33 |
> | 12→19 | 73.58 | 80.30 |
> | 18→20 | 63.88 | 82.93 |
> | 19→2 | 58.59 | 65.85 |
> | 28→20 | 94.76 | 97.56 |
> | 26→2 | 76.72 | 90.24 |
> | 28→2 | 67.28 | 90.24 |
>
> ### Opportunity — SASA vs. Ours (Accuracy, \%)
>
> | Setting | SASA (Acc) | Ours (Acc) |
> |:--:|--:|--:|
> | 1→2 | 81.09 | 86.82 |
> | 1→3 | 76.87 | 86.96 |
> | 1→4 | 82.62 | 84.45 |
> | 2→1 | 77.22 | 89.30 |
> | 2→3 | 75.86 | 84.78 |
> | 2→4 | 82.57 | 81.23 |
> | 3→1 | 84.43 | 83.38 |
> | 3→2 | 67.39 | 81.01 |
> | 3→4 | 80.35 | 82.84 |
> | 4→1 | 88.55 | 93.69 |
> | 4→2 | 84.04 | 82.12 |
> | 4→3 | 85.14 | 88.51 |
>
>
> In summary, from thorough comparison and analysis with prior works, we regard SASA and GCA as complementary and effective under their stated assumptions of stable inter‑variable relationships. Our work targets a different and common wearable MTS setting in which sensor‑wise domain shift violates this invariance, learning a sensor‑level transport and combining it with intra‑domain decorrelation. We have expanded the discussion and citations to make this positioning explicit in our revised manuscript.
>
>
>
> [1] Time Series Domain Adaptation via Sparse Associative Structure Alignment, AAAI 2021.
>
> [2] Transferable Time-Series Forecasting Under Causal Conditional Shift, TPAMI 2024.

---

> ### Author Response · Authors · 2025-11-21
>
> **(Weakness 2, 4 \& Question 1) Definition of Sensor-misalignment and Experiments under Sensor-misalignment**: We thank the reviewer for addressing the point about 'placement or orientation'. We first clarify the term spatial alignment. We defined the term 'spatial misalignment' to indicate distribution shifts that occur mainly due to reattachment of identical sensors, whether it is reattached to a different subject (cross-subject setting as in HAR datasets) or an identical subject multiple times (cross-repetition setting as in sEMG datasets). Accordingly, we believe that the domain discrepancies present in the datasets we evaluate already reflect this form of spatial misalignment, even without any intentional data change. Our empirical results across multiple datasets demonstrate that our method more effectively mitigates these spatial misalignment effects compared to existing baselines. In addition to that, we kindly remind the author that there is an experiment conducted under an intentional sensor permutation setting, where the target sensor configuration is changed, where the features of two sensors in different positions are swapped to each other (6.3 Sensor Permutation Experiment). However, we also acknowledge that our original phrasing may have caused confusion.
>
> Moreover, we thank the reviewer for suggesting a new experimental setting to evaluate performance under heterogeneous sensor configurations, such as different sensor counts. As our method is plug-and-play and most baseline approaches assume identical sensor sets across domains, we simulated heterogeneity by masking a portion of target-domain sensor values (setting them to zero). Across experiments, increasing the fraction of missing sensors from 10\% to 40\% demonstrates that our method remains robust, maintaining performance comparable to the original results. Interestingly, even when 10\% of the target sensors are missing, our method achieves performance comparable to the baseline results obtained with the full sensor set.
>
> Average Performance Under Missing Sensor Ratios on Opportunity HAR dataset.
>
> | Missing Sensor (\%) | Accuracy |
> |--------------------|----------|
> | 10\% | 82.13 |
> | 20\% | 80.79 |
> | 30\% | 69.60 |
> | 40\% | 58.05 |
> | 0 \% (Ours) | 84.68|
> | CLUDA | 82.20|

---

> ### Author Response · Authors · 2025-11-21
>
> **(Weakness 3 \& Question 2) Definition of Sensor**: We first apologize for any confusion caused by the unclear writing. The $N$ corresponds to an individual sensor channel, and we will revise the paper to explicitly define a sensor as a unit comprising a single channel. While this can be viewed as conventional multivariate variable alignment, our method additionally addresses inter-domain spatial alignment, denoted as $\textbf{A}\_{st}$, which aligns representations across domains. Such inter-domain alignment has been largely underexplored in prior work. Figure 2 further illustrates the importance of incorporating this alignment to mitigate distribution shifts in domain adaptation scenarios in multivariate time series datasets. We defined the term sensor that we used throughout the paper in lines 202-204 in our revised paper.

---

### Official Review · Reviewer_8G3n · 2025-10-31

**Soundness:** 2
**Presentation:** 3
**Contribution:** 2
**Rating:** 4
**Confidence:** 5

**Summary:**

The paper proposes IDSA (Inter-Domain Sensor Alignment), a plug-in module for unsupervised domain adaptation (UDA) on wearable multivariate time-series data. Unlike previous approaches focused mainly on simple spatial alignment, IDSA tackles sensor-wise domain shift caused by differences in sensor placement and orientation. It introduces a spatial transportation loss, formulated as a transport problem for inter-domain sensor alignment, and a channel decorrelation loss to reduce redundant intra-domain correlations.

**Strengths:**

1.	IDSA is compatible with existing UDA frameworks such as Deep CORAL and CLUDA, requiring minimal architectural changes and demonstrating performance gains.

2.	The method is theoretically grounded, formulating inter-domain sensor alignment as a transport problem and proving equivalence to the discrete 1-Wasserstein.

3.	IDSA shows strong empirical performance, achieving notable gains on cross-subject HAR and sEMG benchmarks, indicating its effectiveness in addressing sensor-wise domain shifts.

**Weaknesses:**

1.	The evaluation setup in Table 1 seems limited, as the selection of specific source-target domain pairs could be less fair and does not fully test generalization. A fairer and more comprehensive evaluation would be a leave-one-subject-out (LOSO) or leave-one-group-out (LOGO) protocol, where each subject or subject group serves as the target domain in turn. The authors cited some references for this selection, but the selected pairs of this paper differ from those in the referenced papers without a proper explanation.

2.	The proposed decorrelation loss for sparsity is fairly standard and widely used in prior works. While the ablation results indicate that combining it with the transport loss improves performance, it contributes less novelty compared to the main inter-domain sensor alignment component. Also, the connection between the two losses is not clearly explained, making it uncertain whether the observed performance gain results from their genuine interaction or simply from incremental improvements contributed by each loss independently.

3.	My biggest concern lies in the relation of this work to a line of existing works [1,2]. The inter-domain transport formulation shares conceptual similarities with existing methods that align correlation or covariance matrices between the source and target domains of multivariate time-series data, making the degree of novelty somewhat incremental rather than completely novel. For example, [1] also uses sparsity and enforces structural alignment for inter-variable relations.

[1] Time Series Domain Adaptation via Sparse Associative Structure Alignment, AAAI 2021.

[2] Transferable Time-Series Forecasting Under Causal Conditional Shift, TPAMI 2024.

**Questions:**

1.	Regarding the evaluation setup, how were the specific source-target domain pairs in Table 1 selected? Why are they different from the papers cited in section 6.2, paragraph 1?

2.	Concerning the relations of the two loss functions, could the authors clarify the connection between the spatial transport loss and the decorrelation loss? Is there any explicit interaction or dependency between them, beyond their independent contributions? And is there any further evidence for such a connection?

3.	In relation to prior work, have you directly compared IDSA against existing methods that align correlation or covariance matrices between source and target domains of multivariate time-series data (by changing their concept of multi-variate to multi-sensor for your setting)? Beyond the conceptual difference (multivariate to multi-sensor), what are the substantive differences or advantages that distinguish IDSA from these earlier approaches?

---

> ### Author Response · Authors · 2025-11-21
>
> **(Weakness 3 \& Question 3) Comparison Against Existing Methods that Align Correlation or Covariance Matrices (1/2)**:
> We appreciate the reviewer for bringing the references [1][2], which also exploit the relationships across variables (sensors) in domain adaptation. Comparing our work with these works is helpful for positioning our approach, as it shows that the distinction does not stem from a conceptual difference between multi-variable and multi-sensor settings. While these studies and our method all consider relationships between variables (sensors), our problem setting, assumptions, and objectives differ in fundamental ways.
>
>
> | Method          | Domain examples  |   Variable set example |  Task  |
> |-----------------|-----------|--------------|--------|
> | SASA [1]        | Regions, Age-based groups  | (Blood Glucose, Glucagon, Insulin) |   Timeseries UDA |
> | GCA [2]         | Simulations, Regions | (Blood Glucose, Glucagon, Insulin) |  Semi-supervised timeseries forecasting DA |
> | IDSA (Ours)     | Subjects, Repetitions | IMU/sEMG sensor channels (channel 1, channel 2, ..., channel n)  |  Timeseries UDA|
>
> Given a variable set such as (Blood Glucose, Glucagon, Insulin) used in prior works, the variables are well-defined quantities whose correlation or covariance may stably hold across domains.
> Let $X\_1, X\_2, \ldots, X\_n$ denote the variables, and $\varphi$ denotes a feature extractor. In standard DA, one often has
>
> $P\_S\left( y \mid \varphi(X\_1, X\_2, \ldots, X\_n) \right) = P\_T\left( y \mid \varphi(X\_1, X\_2, \ldots, X\_t) \right)$.
>
> However, SASA and GCA further assume that the inter-variable structure is stable:
>
> $P\_S\left( y \mid \varphi(X\_1, X\_2, \ldots, X\_n) \right) \neq P\_T\left( y \mid \varphi(X\_1, X\_2, \ldots, X\_t) \right)$,
>
> $P\_S\left( C \mid \varphi(X\_1, X\_2, \ldots, X\_n) \right) \approx  P\_T\left( C \mid \varphi(X\_1, X\_2, \ldots, X\_n) \right)$,
>  where $C$ denotes their structural dependency.
>
> Under this assumption, their methods learn to extract the invariant structural dependency among variables across different domains, such as:
>
> * Extracting sparse associative structure with intra- and inter-variable attention mechanisms and align those structures [1], and
> * Leveraging domain invariant causal structures while also modeling domain-specific components [2].
>
> We agree that correlation‑ or causality‑based alignment methods such as SASA and GCA are well‑motivated and effective under the assumption that variables are well‑defined and their inter‑variable relationships are approximately stable across domains. However, in wearable MTS, each "variable" is a sensor channel whose semantics can easily change, even if there are no intentional data changes, across subjects or repetitions due to changes in placement from reattachment or characteristics of subjects. This kind of scenario is illustrated in Figure 1 (Right).
> In other words, the wearable MTS domain introduces problem scenarios in which the assumptions underlying methods such as SASA and GCA may not hold.
> We assume that it can result in substantially different variable relations for source and target domains:
>
> $P\_S\left( C \mid \varphi(X\_1, X\_2, \ldots, X\_n) \right) \neq P\_T\left( C \mid \varphi(X\_1, X\_2, \ldots, X\_n) \right)$.
>
> We therefore introduce a channel-wise (variable-wise) transport map $\textbf{A}\_{st}$ and model:
>
> $P\_S\left( C \mid \varphi(X\_1, X\_2, \ldots, X\_n) \right) \approx  P\_T\left( C \mid \varphi(X\_1, X\_2, \ldots, X\_n), \textbf{A}\_{st} \right)$.
>
> With this revised assumption considering 'sensor-wise domain shift', our approach explicitly learns a cross-domain sensor correspondence, in contrast to correlation/covariance alignment that assumes a single shared structure. We model **inter-domain** variable relations, in addition to **intra-domain** variable relations, under the assumption that inter-domain variable relationships may be dynamic.
>
> [1] Time Series Domain Adaptation via Sparse Associative Structure Alignment, AAAI 2021.
>
> [2] Transferable Time-Series Forecasting Under Causal Conditional Shift, TPAMI 2024.

---

> ### Author Response · Authors · 2025-11-21
>
> **(Weakness 3 \& Question 3) Comparison Against Existing Methods that Align Correlation or Covariance Matrices (2/2)**:
> To assess whether the substantial cross-domain differences in variable correlation and covariance require an additional inter-domain variable-wise transformation beyond simple correlation alignment, we conduct the following supplementary experiments on four datasets:Boiler, Air Quality (from prior works), WISDM, and Opportunity (our datasets).
>
> | Dataset | Domains | Variables | Mean Sim. | Std Sim. | Min–Max Sim. | Severity of Cross‑Domain Relation Drift |
> |---|---|---:|---:|---:|---:|---|
> | Opportunity | 4 subjects  | 113 | 44.89 | 11.57 | 35.44–69.83 | Severe  |
> | WISDM  | 30 subjects  | 3 | 66.61 | 18.71 | 40.19–88.06 | High variance |
> | Air quality | 4 cities  | 11 | 81.29 | 7.91 | 67.49–91.75 | Moderate (local conditions) |
> | Boiler | 3 conditions  | 20 | 94.38 | ~2.7 | 92.47–97.84| Benign (stable physical sensors) |
>
> For each domain, we aggregate channel-wise time-series into feature statistics (mean/std) and apply PCA to stabilize dimensionality and compute inter-variable correlation matrix from this. Domain similarity inside a dataset is then defined as the cosine similarity between vectorized correlation matrices. Corresponding domain similarity heatmaps of all four datasets is available in Appendix of the revised manuscript.
> These results show that, in the cross‑subject / cross‑repetition setting, wearable datasets (Opportunity, WISDM) exhibit substantial drift in inter‑variable (sensor) relationships, whereas non‑wearable datasets (Air, Boiler) with well-defined variable set are more stable. In particular, Opportunity shows low mean similarity across domains (≈45\%) and WISDM shows the second lowest with highly variable similarity (≈67\% with a large spread), indicating that a single shared correlation/covariance structure is often insufficient. By contrast, Air quality (≈81\%) and Boiler (≈94\%) display higher cross‑domain similarity. These observations motivate the need for an additional inter‑domain transportation process at the variable (sensor) level rather than relying solely on correlation alignment that assumes a single invariant structure. The comprehensive results of this experiment, including experimental setup, visualization, and more thorough analyses, are now provided in **Appendix A.7.** of the revised paper.
>
> We also evaluated SASA on WISDM and Opportunity datasets under the same scenarios used in our work for a direct comparison with IDSA. Our method  outperformed SASA in accuracy across both datasets. We will incorporate these comparison results, including F1 scores, into our revised manuscript.
>
> ### WISDM — SASA vs. Ours (Accuracy, \%)
>
> | Setting | SASA (Acc) | Ours (Acc) |
> |:--:|--:|--:|
> | 2→28 | 84.94 | 86.67 |
> | 7→2  | 71.25 | 82.93 |
> | 7→26 | 81.08 | 78.05 |
> | 12→7 | 67.30 | 83.33 |
> | 12→19 | 73.58 | 80.30 |
> | 18→20 | 63.88 | 82.93 |
> | 19→2 | 58.59 | 65.85 |
> | 28→20 | 94.76 | 97.56 |
> | 26→2 | 76.72 | 90.24 |
> | 28→2 | 67.28 | 90.24 |
>
> ### Opportunity — SASA vs. Ours (Accuracy, \%)
>
> | Setting | SASA (Acc) | Ours (Acc) |
> |:--:|--:|--:|
> | 1→2 | 81.09 | 86.82 |
> | 1→3 | 76.87 | 86.96 |
> | 1→4 | 82.62 | 84.45 |
> | 2→1 | 77.22 | 89.30 |
> | 2→3 | 75.86 | 84.78 |
> | 2→4 | 82.57 | 81.23 |
> | 3→1 | 84.43 | 83.38 |
> | 3→2 | 67.39 | 81.01 |
> | 3→4 | 80.35 | 82.84 |
> | 4→1 | 88.55 | 93.69 |
> | 4→2 | 84.04 | 82.12 |
> | 4→3 | 85.14 | 88.51 |
>
>
> In summary, from thorough comparison and analysis with prior works, we regard SASA and GCA as complementary and effective under their stated assumptions of stable inter‑variable relationships. Our work targets a different and common wearable MTS setting in which sensor‑wise domain shift violates this invariance, learning a sensor‑level transport and combining it with intra‑domain decorrelation. We have expanded the discussion and citations to make this positioning explicit in our revised manuscript.
>
>
>
> [1] Time Series Domain Adaptation via Sparse Associative Structure Alignment, AAAI 2021.
>
> [2] Transferable Time-Series Forecasting Under Causal Conditional Shift, TPAMI 2024.

---

> ### Author Response · Authors · 2025-11-21
>
> **(Weakness 1 \& Question 1) Clarification of Table 1 Evaluation Setup  and Additional LOSO-Based Comprehensive Analysis:**
> We apologize for any confusion caused by the unclear explanation of the evaluation setup in Table 1, particularly regarding its correspondence to the prior works cited in section 6.2, as the reviewer mentioned.
>
> Our intention was to clarify that the evaluation protocol follows the same scheme adopted in the cited studies, wherein ten scenarios are randomly selected. The specific configuration of these scenarios matches that of the first referenced work [3]. To clarify this point explicitly, we have revised the corresponding part in the revised version.
> Furthermore, we conduct additional LOSO-based analysis for more comprehensive analysis on the WISDM dataset, as the reviewer suggested, by comparing with the Deep Coral and CLUDA methods with and without applying our method. Because the LOSO setting yields a large number of source domains, most scenarios achieve 100\% accuracy on all 4 methods. The results of non-100\% scenarios are shown in the following table. The result indicates that the proposed method outperforms when incorporated into existing baselines.
> Interestingly, after applying PCA to the raw features, the target domain in the table (non-100\% scenarios) shows very low similarity to the other subjects, as detailed in the Appendix Figure of the revised manuscript.
>
> | Scenario (Target domain)          | 10    | 18    | 27   | 28    | 30    |
> |-----------------|-------|-------|------|-------|-------|
> | Deep Coral              | 81.08 | 92.45 | 88.00   | 91.11 | 88.46 |
> | ours + Deep Coral       | 81.08 | 92.45 | 92.00   | 93.33 | 88.46 |
> | CLUDA           | 70.27 | 88.68 | 96.00   | 88.89 | 84.62 |
> | ours + CLUDA   | 70.27 | 92.45 | 100.0  | 91.11 | 86.54 |
>
>
> [3] CLUDA: Contrastive Learning for Unsupervised Domain Adaptation of Time Series, ICLR 2023.

---

> ### Author Response · Authors · 2025-11-21
>
> **(Weakness 2 \& Question 2) Contribution of the Decorrelation Loss:**
> Thanks for pointing out an interesting observation. We also acknowledge that the proposed decorrelation loss is widely used in prior works. We want to highlight that there is a synergy between the two loss functions under the assumption that the source data instance and target data instance have identical classes. Under this assumption, the synergy between the decorrelation loss and our proposed loss can be interpreted through an information-bottleneck perspective.
>
> > [Theorem] Let the source-wise information bottleneck be defined as: ${IB}\_{s} = I(\textbf{Z}\_{t}, {X}\_{s}) - \beta I(Tar, \textbf{Z}\_{t}),$ where ${X}\_s$ indicates the source distribution and $Tar$ denotes target domain-specific information. In particular, if IDSA is optimized through loss functions, the transformed target representation satisfies the source-wise information bottleneck. $\min ({L}\_{\text{st}} + {L}\_{\text{dec}}) \Rightarrow \max {IB}\_{s}.$
>
> Our transport plan enforces each target instance to align its sensor-wise distribution with that of the corresponding source instance, effectively encouraging $ \textbf{Z}\_{s} \approx \textbf{Z}\_t $.
> When combined with the decorrelation loss, this objective promotes target representations $\textbf{Z}\_t$ that preserve high mutual information with the source distribution while suppressing mutual information with target-specific domain factors $Tar$. Consequently, the learned representation becomes well-suited for domain adaptation. We appreciate the reviewer’s insightful comment, and the theoretical discussion is incorporated as Theorem 2, and additional discussion is in Appendix A.2.3 in our revised paper.

---

### Author Response · Authors · 2025-12-03

## Follow-up discussions summarization:
After the first rebuttal round as above, reviewers **[vQxH, E2qa]** additionally raised the following questions:

- Clarification of comparison setting and results for UniMTS: possible additional sensor modalities **[vQxH]**, not fully convincing results (mixed on WISDM, unfair setting on Opportunity) **[E2qa]**
    - We clarified that UniMTS is architecturally tailored to IMU sensor–channel structure in its embedding equations, so incorporating non‑IMU sensor modality would require non‑trivial architectural changes.
    - We additionally analyzed the comparison setting for fairness clarification, pointing out that UniMTS already benefits from an additional textual modality in its pre-training protocol, as summarized in Table a. **(Section 6.2.)**
    - We further conducted additional experiments on Opportunity with a more controlled setting (IMU sensors only for ours), and still observed overall performance gains for ours. **(Appendix)**

#### Table a. Experimental setting for UniMTS and ours
| Dataset                                | UniMTS Input & Setting                                          | Ours Input & Setting                                  |
|----------------------------------------|------------------------------------------------------------------|--------------------------------------------------------|
| **WISDM** (only IMU sensors)                   | Full sensor channels + **textual descriptions** | Full sensor channels            |
| **Opportunity** (heterogeneous sensors) – Original | IMU sensor channels only + **textual descriptions** | Full heterogenous sensor channels  |
| **Opportunity** (heterogeneous sensors) – Controlled | IMU sensor channels only + **textual descriptions** | IMU sensor channels only

---

### Author Response · Authors · 2025-12-03

# Rebuttal Summary to AC
We sincerely thank all reviewers for their thoughtful feedback and insightful suggestions. During the rebuttal process, these points have been **explicitly taken into account** and **incorporated into the revised manuscript**, either as new content, clarifications, or further elaborations to improve both the clarity and technical depth of our work. We indicate the revised parts of the paper as **(sections)** grouping raised concerns for brevity.

## Initial review summarization:
### Reviewers commented on the strengths as follows:
- **[G3n, vQxH, E2qa]** The paper presents strong empirical results, showing large performance gains over baselines.
- **[8G3n, Ly5z]** The paper presents a practical and adaptable design that can be integrated into existing UDA frameworks (plug and play module).
- **[Ly5z, vQxH]** The research question is interesting and meaningful, which could have broad applications in the real-world.
- **[Ly5z, vQxH, 8G3n]** The paper has an understandable, clear presentation including intuitive figures and mathematical derivations with grounded theoretical analysis.
- **[E2qa]** Well-investigated reasonable motivation with a definition of sensor-wise domain shift.
- **[E2qa]** The proposed method is aligned with the motivation.

### The questions raised in the initial review and our subsequent efforts in response:
- **[8G3n, Ly5z]** Novelty concerns due to missing comparisons with existing methods that align correlation or covariance, such as SASA or GCA
    - We provided a detailed conceptual comparison analysis with SASA and GCA, showing that their key assumptions may not hold under sensor-wise distribution shift, where variable relationships across domains are dynamic. **(Section 2, 3, Appendix)**
    - We evaluated SASA as an additional baseline on WISDM and Opportunity, showing overall performance gain of ours. **(Section 6.2)**

- **[E2qa]** Novelty concerns due to missing baselines of existing OT-based methods, such as MLOT, OTDA, or DAVES
    - We clarified that RAINCOAT is already an OT-based baseline. **(Appendix)**
    - We added comparison analysis across 4 OT-based methods (MLOT, OTDA, DAVES, RAINCOAT) and ours, showing fundamental differences. **(Appendix)**
    - We conducted an experiment with MLOT and OTDA on WISDM and Opportunity, showing overall performance gain of ours. **(Appendix)**

- **[Ly5z]** Definition and empirical justification for sensor-misalignment
    - We clarified that the spatial misalignment naturally occurs across domains even without intentional sensor-wise changes, consistent with our cross-subject (Table 1) and cross-repetition (Table 2) empirical settings. **(Section 1)**
    - We clarified that the paper already included an experiment (Figure 5) under a non-identical sensor deployment.
    - We conducted an experiment with a more comprehensive non-identical sensor deployment scenarios by randomly removing sensors and provided result analysis.
    - We added visualizations of the transportation map including sensor-wise swap scenario illustrating robustness of our method under sensor-misalignment. **(Section 6.3)**

- **[8G3n, E2qa]** Contribution of the decorrelation loss
    - We incorporated a new theorem that applying two loss functions together has a synergistic effect, which we connect with the information bottleneck theorem, elaborating theoretical analysis. **(Section 5.4)**

- **[8G3n]** Evaluation setup in Table 1 and the suggestion of LOSO setting
    - We clarified that our evaluation protocol follows one adopted in previous studies. **(Section 6.2)**
    - We experimented under the suggested LOSO, showing that ours outperforms in overall.

- **[vQxH, E2qa]** Lack of recent MTS baselines, such as UniMTS and CrossHAR
    - We added comparison with two suggested methods (UniMTS, CrossHAR).
    - We evaluated UniMTS on WISDM and Opportunity as an additional baseline and reported results analysis. **(Section 6.2)**

- **[E2qa]** Missing visualizations of transportation map and correlation matrix
    - We added visualizations of the transportation map, illustrating the robustness of the transportation map **(Section 6.3)**
    - We added visualizations of self-correlation heatmaps. **(Appendix)**

- **[Ly5z, vQxH]** Clarity of the terminology "sensor" and writing in the abstract/introduction.
    - We clarified that in our paper a “sensor” means a single channel, and an IMU unit consists of multiple such channels. **(Section 4.1)**
    - We revised the abstract and introduction in line with the reviewer's suggestion. **(Abstract, Section 1)**

- **[vQxH]** Setting and interpretation of Figure 2 and Opportunity dataset
    - We clarified that the similarity used in Figure 2 is computed as cosine similarity between sensor channels.
    - We clarified and elaborated the notion of “functional consistency”.
    - We added a precise description of the Opportunity dataset setting. **(Appendix)**

---

### Meta-Review · Area_Chair_JStV · 2026-01-07

**Summary:**

This paper presents IDSA (Inter-Domain Sensor Alignment), a plug-in module for unsupervised domain adaptation of wearable multivariate time series. IDSA introduces a sensor-level optimal-transport objective that learns a cross-sensor transport map to realign target channels with source channels, together with a channel-decorrelation regularizer to suppress spurious intra-domain couplings. The method is theoretically motivated (sensor-level discrete Wasserstein equivalence) and designed to be added to existing UDA backbones. The authors evaluate IDSA across multiple HAR and sEMG benchmarks and have performed extensive experiments.

The authors engaged substantively with reviewer concerns, added missing baselines and stress tests, provided clarifications and additional theory, and demonstrated practical, reproducible gains. It fills an important practical gap for wearable MTS and offers a useful, well-validated approach for the community.

However the presentation and the mixed experiment results mean that the paper has not pushed itself beyond the borderline territory.

**Reviewer Concerns:**

After reading through the extensive rebuttal, the remaining objections are mainly about incremental novelty and experiment results. This is a subjective reviewer judgment.
The main concern that remains open is the mixed performance outcomes.
Although the authors have written also about the potential unfair comparison setting, this has not been further clarified in the revision, as the experiment was done in a very strict setting.

**Reviewer Scores:**

I think it is likely the reviewer with 6 would have increased their scores. However those with the 2s have said they won't increase their score and the one with 4 hasn't been responsive in the early stage.

---

### Decision · Program_Chairs · 2026-01-26

Reject